# Neuronal baseline shifts underlying boundary setting during free recall

Yitzhak Norman[1], Erin M. Yeagle[2], Michal Harel[1], Ashesh D. Mehta[2] & Rafael Malach[1]

Asked to freely recall items from a predefined set (e.g., animals), we rarely recall a wrong exemplar (e.g., a vegetable). This capability is so powerful and effortless that it is essentially taken for granted, yet, surprisingly, the underlying neuronal mechanisms are unknown. Here we investigate this boundary setting mechanism using intracranial recordings (ECoG), in 12 patients undergoing epilepsy monitoring engaged in episodic free recall. After viewing vivid photographs from two categories (famous faces and places), patients were asked to freely recall these items, targeting each category in separate blocks. Our results reveal a rapid and sustained rise in neuronal activity ("baseline shift") in high-order visual areas that persists throughout the free recall period and reflects the targeted category. We further show a more transient reactivation linked to individual recall events. The results point to baseline shift as a flexible top–down mechanism that biases spontaneous recall to remain within the required categorical boundaries.

[1] Department of Neurobiology, Weizmann Institute of Science, 76100 Rehovot, Israel. [2] Department of Neurosurgery, Donald and Barbara Zucker School of Medicine at Hofstra/Northwell, and Feinstein Institute for Medical Research, Manhasset, NY 11030, USA. Correspondence and requests for materials should be addressed to R.M. (email: rafi.malach@gmail.com)

Asked to freely recall a list of items from a specified category (e.g., high-school classmates), individuals may encounter difficulty in recalling certain exemplars, but they will rarely recall a wrong exemplar (e.g., their grandmother). Preventing such errors, also termed intrusions[1], is so efficient and easy[2,3] that we rarely even notice this powerful cognitive ability. However, despite its robustness and ubiquitous nature, the neuronal mechanism that allows such categorical constraining is still unknown. Previous neuroscientific studies that addressed self-initiated recall have mainly focused on the conditions and mechanisms underlying the spontaneous emergence of specific recall events, rather than the voluntary top–down control mechanism that limits the free retrieval to specific categories or sets. Thus, studying free recall of previously shown video clips, Gelbard-Sagiv et al.[4] were able to demonstrate a content-specific reactivation of medial temporal lobe (MTL) neurons that anticipated the individual recall events by 1–2 s. However, it is important to emphasize that the spontaneous recall of specific memory items and the top-down limitation of such spontaneous recalls within specific category bounds are two separate, and in fact complementary, processes. So far, the neuronal mechanism that allows constraining memory retrieval to a predefined bounded set is unknown.

To examine this top–down boundary setting mechanism in a way that separates it from the individual events of specific item recall, it is critical to use a method that has sufficient temporal

**Fig. 1** Experimental design, stimuli, and electrode locations. **a** Participants viewed pictures of famous people and places, and after a short distraction task, were instructed to freely recall items from each category at a time. **b** Example of pictures from the two categories. Participants were instructed to remember the pictures in as much detail as possible, focusing on colors, face expressions, perspective, and so on. Photo of Bill Clinton courtesy of Gage Skidmore; photo of the Golden Gate Bridge courtesy of Nicolas Raymond; photo of the Leaning Tower of Pisa courtesy of Josu. These pictures are all published under a Creative Commons license. Photo of Barack Obama courtesy of the US Government. **c** Multi-patient electrode coverage. Recording sites in each patient were localized based on post-operative CT and MRI scans; the cortical surface was then reconstructed using FreeSurfer and standardized using SUMA to allow visualization of electrodes from different patients on a single cortical template (see Methods section). Visually responsive electrodes were attributed based on anatomical and functional criteria to one of the following subgroups: V1 (blue); V2 (light blue); category non-selective intermediate visual areas (yellow); face selective (red); and place selective (green). Note the clear tendency of category selective electrodes to be localized in high-order visual areas along the ventral stream. See Supplementary Table 2 for further details. (Abbreviations: RH—right hemisphere; LH—left hemisphere)

resolution to enable the temporal distinctions between these two types of processes. Furthermore, it is important to employ a methodology with both high temporal resolution and sufficient spatial selectivity to differentiate between neuronal representations of different contents. We therefore approached this question using intracranial ECoG recordings in patients while they engaged in free recall of visual images constrained to particular categories. We and others have previously demonstrated that such recordings show exquisite content selectivity[5–8], which is robustly linked to the average spiking activity in the recording sites when examining the power of high-frequency broadband (HFB, 60–160 Hz) activity[9–11].

Our results reveal that, during free recall, high-order visual representations show a sustained goal-directed enhancement of neuronal activity, which we have coined "baseline shifts". These baseline shifts match the category selectivity of these representations. Importantly, using a number of control analyses we demonstrate that these baseline shifts constitute a separate process from that of the specific recall events. These findings are in line with previous work, which has demonstrated a similar modulation of baseline activity induced by top–down attention during anticipation of external visual stimuli[12–15].

Taken together, our findings point to a powerful and flexible top–down mechanism that allows spontaneous recall to remain within the required boundaries. This neuronal mechanism is compatible with different cognitive models of free recall[16–20], as well as with recent ideas about the role of top–down attention in strategic control of retrieval processes[21, 22].

## Results

**Task overview and behavioral performance.** Twelve epilepsy patients participated in the study, each implanted with 105–256 subdural electrodes as part of a clinical monitoring procedure for diagnostic purposes (see Supplementary Table 1 for demographic details and distribution of electrodes across individuals). ECoG activity was recorded during rest, and while patients performed a visual free recall task. The specific details of the task are depicted in Fig. 1a (see Methods section for further details). Briefly, the task consisted of two runs, each beginning with a resting-state period that served as a baseline for the free recall session. Following that, participants viewed 14 different pictures of famous faces and popular landmarks, with four repetitions of each exemplar. After a short interference task (backward counting), the patients were asked to freely recall these pictures, targeting each category (faces/places) in separate blocks. We instructed the patients to not only name but also to describe each picture they recall with 2–3 prominent visual features. This was done to ensure that the patients also retrieved visual information specific to the studied items, rather than just general semantic details. The order of recalled categories was fixed across patients and counter-balanced between the two runs (starting with faces in the first run and places in the second run).

On average, participants recalled 8.79 (±0.6) items per run, and had 12.8 (±1.22) 'recall events' per run when including recurring recollections (numbers in parentheses indicate SEM). Recall events were defined as any verbal utterance in which participants began to describe a specific picture (see Methods section). Recall performance was similar between the two categories (average number of recalled items per run: 4.38 faces, 4.42 places; see Supplementary Fig. 1 for individual subject recall performance and distribution of recall events across time). The mean accuracy on the task (i.e., correct recalls) was 88.49% (±2.18%). The remaining 11.51% were 'extra-category intrusions', i.e., recall events in which the patients erroneously retrieved an item from the non-designated category. Such intrusion rate is considered

high compared to previous reports[2]; however, this discrepancy most likely reflects the fact that unlike previous studies, here we specifically instructed the patients to say everything that came to mind during the free recall period.

**Visually responsive electrodes.** The analyses of ECoG data was focused on changes in high-frequency broadband amplitude (HFB: 60–160 Hz, also known as high-gamma), which was shown to be an excellent marker of local neuronal population activity[9–11]. During the presentation of the pictures visual electrodes showed highly robust and consistent HFB responses ($p < 0.01$ FDR corrected, paired $t$-test, pre-stimulus baseline vs. post-stimulus response, see Methods section). An overall view of electrodes' location and their relationship to different visual areas is presented in Fig. 1c. Visually responsive electrodes were localized to posterior cortical regions, including V1, V2, intermediate visual areas, and high-order visual areas in the inferior temporal and the lateral-occipital cortices (see Supplementary Table 2 for MNI coordinates and further details). In agreement with previous studies reviewed in Grill-Spector and Weiner[23], face-preferring and place-preferring electrodes were typically localized laterally and medially to the fusiform gyrus, respectively.

For convenience, we will use the term "face-selective" and "place-selective" for visual electrodes that showed significantly higher HFB activation ($p < 0.01$, FDR corrected) to faces and places, respectively, during the picture viewing stage of the task (Methods section). For analysis, we grouped the remaining visual electrodes depending on their anatomical location and response latency in three additional subgroups: V1 electrodes, V2 electrodes, and intermediate visual areas (Fig. 1; see Methods section, Supplementary Tables 1, 2 for further details).

**Content-specific baseline shifts during free recall.** For clarity, we first present data from two representative category-selective visual electrodes. Following these examples, we present a comprehensive statistical analysis of the group results.

Examining the activity profiles of face-selective and place-selective electrodes during the free recall of faces and places revealed a small but highly consistent shift in overall HFB activity starting immediately following the cue to recall a specific category of images. Importantly, this change in baseline activity was specific to the image category that the patients were instructed to recall. An example of this "baseline-shift" effect in two representative high-order visual electrodes is shown in Fig. 2. During the picture viewing stage the two electrodes manifested a significant category-selective response ($p < 0.01$ FDR corrected, two sample $t$-test; see Methods section) (Fig. 2b). The first electrode, which exhibited selectivity to faces, was located on the lateral portion of the fusiform gyrus (also known as FFA-1)[23]. The second electrode, which exhibited selectivity to places, was located medial to the fusiform gyrus at the lateral edge of the collateral sulcus (CoS) (Fig. 2c). During the free recall period, these two electrodes exhibited a selective increase in baseline HFB amplitude when the patient attempted to recall images from their preferred category (change in median amplitude: face-selective electrode, 12.23%; place-selective electrode, 8.28%) (Fig. 2a).

To examine whether this category-selective baseline-shift was a general phenomenon across the entire group of high-order category-selective electrodes, we normalized the HFB amplitude in each electrode by the geometric mean amplitude during resting state, recorded at the beginning of each run (Methods section), and computed the median of the normalized HFB amplitude during the entire free recall period, separately for the face recall

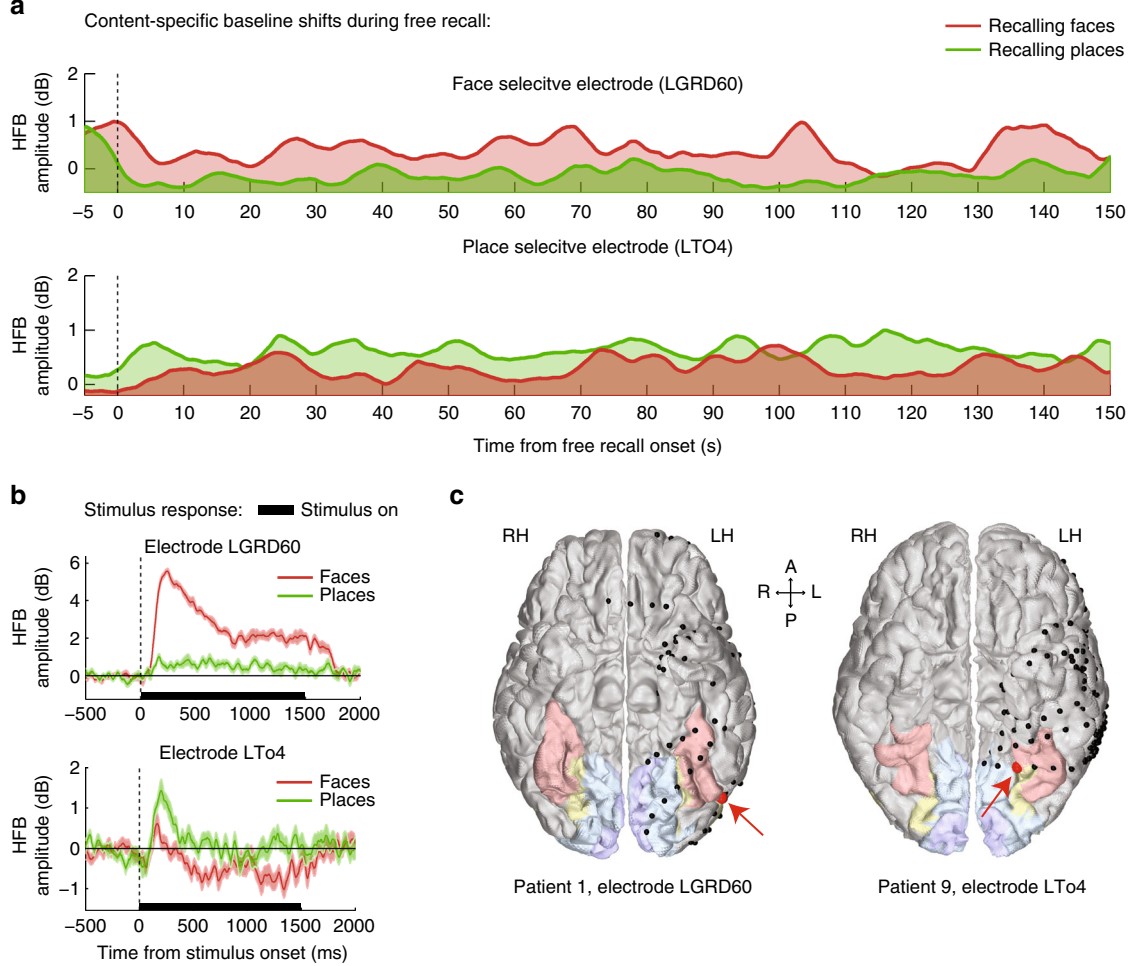

**Fig. 2** Baseline shifts during free recall in two representative electrodes. **a** An example of a face-selective electrode (top) and a place-selective electrode (bottom) showing a sustained baseline increase during free recall of pictures from their preferred category. HFB time-courses were normalized relative to rest and smoothed using a triangular window of 5 s for visualization purposes. **b** Stimulus response of the two electrodes during the picture viewing stage, showing a clear preference to either faces or places (shaded areas represent SEM). **c** Anatomical locations of the two electrodes (marked in red) in relation to primary visual cortex (blue), intermediate visual areas (yellow), and the fusiform gyrus (pink)

and the place-recall periods (150 s each). The median amplitude was averaged across the two runs of the task.

A two-way mixed-design ANOVA with the fixed factors of 'electrode group' and 'recalled category' and the random factors of 'patient' and 'electrode' (see Methods section for further details), revealed a highly significant interaction ($F(1,63) = 17.69$, $p < 0.0001$), indicating that each group of electrodes increased its HFB amplitude when the patients tried to recall images from its preferred category (Fig. 3a). An additional Wilcoxon signed-rank test indicated that median HFB amplitude during free recall was higher than the resting-state baseline ($Z = 5.29$, $p < 10^{-6}$, $n = 65$ electrodes). We found a similar general increase in median HFB amplitude relative to rest in V2 electrodes, but not in V1 (free recall compared to rest, V2: $p = 0.0028$, V1: $p = 0.46$, Wilcoxon signed-rank tests; Supplementary Fig. 3a).

Additional analysis of within electrode differences, subtracting the median amplitude during recall of the preferred category from the median amplitude during recall of the non-preferred category similarly revealed a significant increase in median amplitude during recall of the preferred category, indicated by a shift of the entire distribution of differences to the right (mean amplitude increase: 1.25%, interquartile range (Q1–Q3): −0.1 to 2.6%, $p < 0.0001$, Wilcoxon signed-rank test; see Fig. 3b, and Supplementary Fig. 4b for a spaghetti plot). A one-way repeated

measures ANOVA comparing early (0–50 s), middle (50–100 s), and late (100–150 s) stages of free recall found no significant differences between stages ($F(2,128) = 1.94$, $p = 0.15$), and a Wilcoxon signed-rank test indicated that the amplitude gain was significantly positive throughout the three stages (early: $Z = 4.04$, $p < 10^{-4}$; middle: $Z = 4.06$, $p < 10^{-4}$; late: $Z = 2.26$, $p = 0.02$; Wilcoxon signed-rank test, $n = 65$ electrodes), suggesting a rather steady amplitude gain (Fig. 3c).

Next we examined whether the magnitude of the baseline-shift was related to the level of category selectivity of each electrode. To this end, we first constructed a category selectivity index (CSI) that measured the strength of category selectivity during the viewing stage:

$$\text{CSI} = \frac{\overline{R}_{\text{faces}} - \overline{R}_{\text{places}}}{\overline{R}_{\text{faces}} + \overline{R}_{\text{places}}},$$

where $\overline{R}_{\text{faces}}$ and $\overline{R}_{\text{places}}$ are the mean stimulus response in each category, averaged over a time window of 100–500 ms post-stimulus and normalized by the maximal stimulus response in each electrode. Selectivity values can vary from −1.0 to 1.0, with positive values indicating preference to faces, and negative values indicating preference to places.

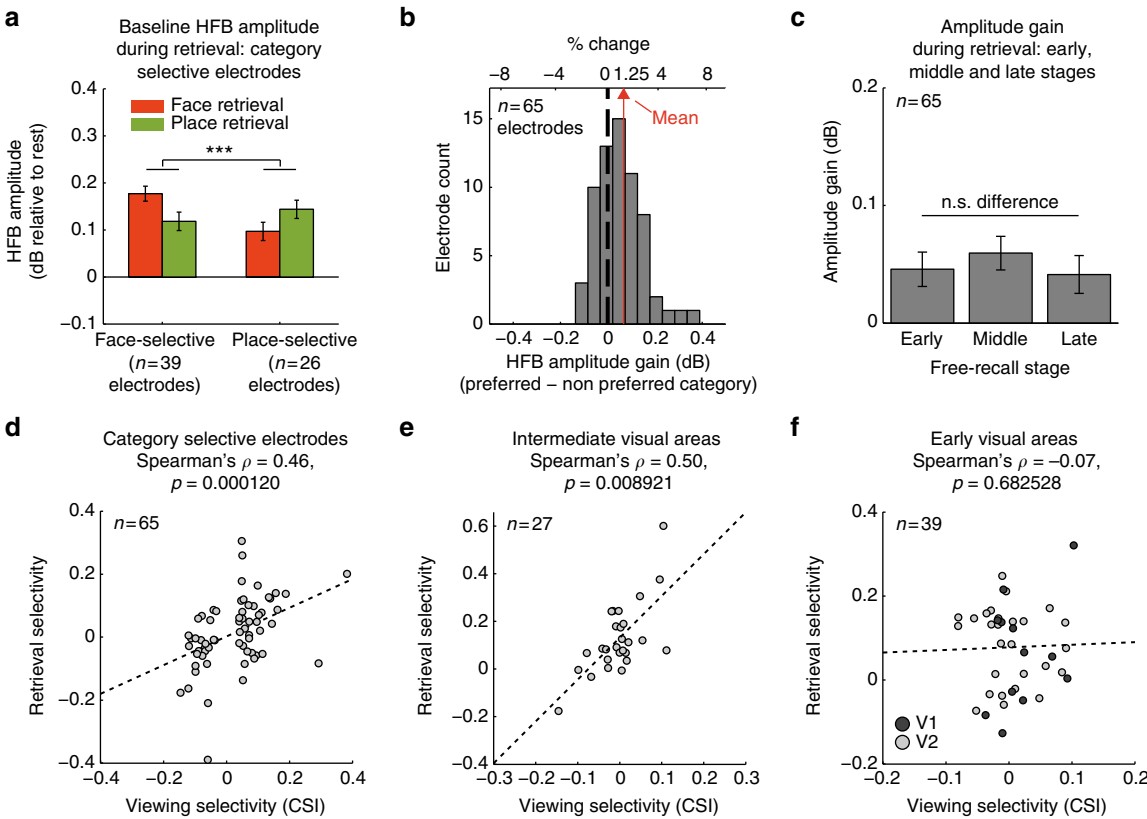

**Fig. 3** Content-selective baseline shifts during free recall—group results. **a** Median HFB amplitude in category selective electrodes was increased in a content-selective manner throughout the free recall period (two-way mixed-effects ANOVA, interaction: $F(1,63) = 17.69$, ***$p < 0.0001$). **b** A histogram showing HFB amplitude gain during recall across all category selective electrodes: preferred minus non-preferred category, $p < 0.0001$, signed-rank test. **c** HFB amplitude gain during early, middle and late recall stages: preferred minus non-preferred category. There was no significant difference between the three stages ($p = 0.15$, one-way repeated measures ANOVA). **d–f** Correlation between category-selectivity index (CSI) during viewing and baseline-shift selectivity during recall (all category selective electrodes: Spearman's $\rho = 0.46$, $p = 0.0001$; intermediate visual areas: Spearman's $\rho = 0.50$, $p = 0.009$; V1 and V2: Spearman's $\rho = -0.07$, $p > 0.68$). In contrast to high visual areas, early visual electrodes did not show any correlation. Error bars represent SEM

We then quantified the strength of selectivity during recall by measuring the difference between the median HFB amplitude during face recall and place recall, with positive values indicating greater baseline activity during face recall, and negative values indicating greater baseline activity during place recall.

Correlation between these two independent selectivity measures would indicate that category preference during viewing was preserved during free recall, and that the magnitude of baseline shifts corresponded to the magnitude of category selectivity. Indeed, visually responsive electrodes located in intermediate visual areas, as well as the entire group of high-order category selective electrodes, manifested highly significant correlation (Fig. 3d, e: spearman's $\rho = 0.50$, $p = 0.009$, all category selective electrodes: spearman's $\rho = 0.46$, $p = 0.0001$). Conversely, attempting to search for a link between category selectivity and baseline-shift in early visual cortex failed to show a significant effect (see Fig. 3f and Supplementary Fig. 3b, c)—perhaps related to the very weak category selectivity of electrodes in these areas during picture viewing.

To test the reliability of the baseline shift phenomenon across the two runs of the task, we carried out a follow-up analysis in which we extended the two-way mixed-effects model described in Fig. 3a into a three-way model with fixed factors of 'electrode group', 'recalled category' and 'run', and random factors of 'patient' and 'electrode'. To allow a more adequate comparison between runs, prior to the ANOVA the HFB amplitude in each electrode was normalized by the mean amplitude across the entire

free recall session in each run, which minimized variability related to resting state differences between runs and electrodes.

We found again the same significant interaction between electrode group and recalled category ($F(1,189) = 24.94$, $p < 0.0001$), with neither run effect nor interaction between run, electrode group and recalled category) $F(1,189) = 0.11$, $p = 0.74$). We did find, however, a significant interaction between run and recalled category ($F(1,189) = 11.1$, $p = 0.001$), which was expected given that the order of recalled categories was counter balanced between the two runs. Such interaction pointed to a possible recall-order effect, i.e., the effect of being targeted first or second. Therefore, we carried out an additional mixed-design ANOVA, this time using the fixed factors of 'electrode group', 'recalled category' and 'recall order' (and the same random factors). Here again the interaction between recalled category and electrode group was highly significant ($F(1,189) = 24.94$, $p < 0.0001$). However, we also found a significant recall order effect ($F(1,189) = 11.1$, $p = 0.001$) with no further interaction (Supplementary Fig. 4a). A direct comparison between the two levels of recall-order (targeted first vs. targeted second) revealed an overall increase in HFB activity for both preferred and non-preferred categories when targeted second (i.e., after the opposed category was already retrieved) ($t(189) = -3.33$, $p = 0.001$, mixed-effects estimate of amplitude increase: 0.04 dB).

**Sustained baseline shifts rather than transient reactivations.** While our analysis revealed a significant category-specific change

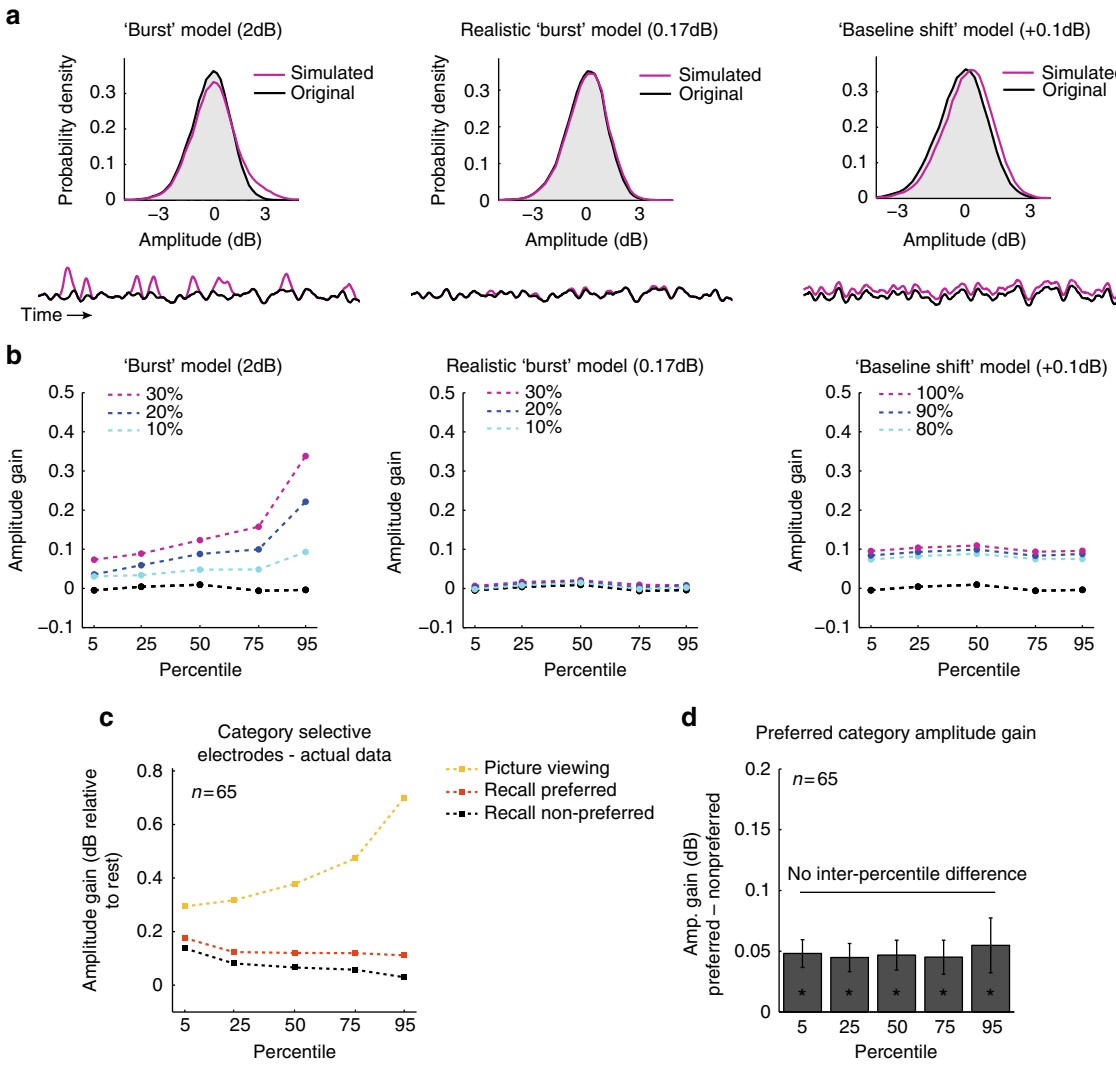

**Fig. 4** Amplitude increase during retrieval reflects baseline-shift rather than transient bursts. **a** Top: an example log-normal distribution of HFB amplitude values in a representative face-selective electrode during retrieval of the non-preferred category (black). Magenta lines illustrate distributions of simulated signals (shown below) that represent the three main alternatives for the underlying process: 'burst' model—in which we artificially inserted sporadic bursts of 2 dB (convolved with a 3 s Gaussian window) to 10% (light blue) 20% (blue) and 30% (magenta) of the signal to simulate a clear case of burst dynamics; a more realistic "burst" model in which we artificially inserted sporadic bursts of 0.17 dB to 10, 20, or 30% of the signal to simulate recall-related activations similar to the response during individual recall events reported in Fig. 5; and finally, a 'baseline shift' model, in which we added a constant gain of 0.1 dB to 80, 90, or 100% of the signal. **b** Differences between the three models are evident when comparing amplitude changes across percentiles. As can be seen, only the 'baseline-shift' model leads to a constant gain across all percentiles. **c, d** In the actual data, category selective electrodes manifest statistical properties that are more compatible with the baseline-shift model: a significant amplitude gain across all percentiles during recall of the preferred category (*$p < 0.001$, preferred minus non-preferred category, signed-rank test), with no significant inter-percentile differences ($p > 0.84$, Kruskal–Wallis test). For comparison, during stimuli presentation the same electrodes show an amplitude modulation more compatible with a "burst" model rather than baseline-shifts, as expected when presenting transient stimuli (yellow). Error bars represent SEM

in median HFB amplitude, it could be argued that this change did not constitute a true stable baseline-shift, but rather an artifact due to accumulation of many transient recall events over a long time window (150 s).

To examine this possibility, we carried out two control analyses. First, based on the patients' voice recordings (see Methods section), we extracted all periods of overt recollection (from speech onset to offset) and recomputed the median HFB amplitude while excluding data points from 2 s before the onset until the offset of each individual recall. A mixed-design ANOVA, identical to the one reported in Fig. 3a, indicated again a highly significant interaction effect (F(1,63) = 7.91, p = 0.007), only this time the effect was entirely due to activity during the

inter-recall intervals, when patients were involved in memory search rather than in an overt recollection.

Second, we looked at the statistical distribution of momentary HFB amplitude levels during the free recall period. A situation where the activity was dominated by many separate individual recall events should lead to a very different distribution of amplitude values compared to the situation where there was a stable overall baseline-shift (Fig. 4).

We first considered in a simple model how the distribution of HFB amplitude values would look if we assume no baseline-shift so that the entire response dynamics consisted of several independent bursts of activations linked to individual recall events (a "burst" model). We explored two versions of this model. The first was aimed to illustrate a clear case of sporadic burst

dynamics. It was constructed by adding sporadic bursts of 2 dB (convolved with a 3 s Gaussian window) to a typical HFB signal taken from a representative face-selective electrode during recall of places. The second simulation was constructed by adding to the same representative signal sporadic bursts of 0.17 dB, to simulate a more realistic burst amplitude, given the observed response to individual recall events presented below (peak amplitude difference between preferred and non-preferred images, Fig. 5b).

Bursts frequency was explored at 10, 20, and 30% of the total duration of the free recall period.

We also considered the alternative case in which the activation change observed during free recall was solely due to a constant overall change in baseline activity (i.e., a constant gain of 0.1 dB) that persisted at a fairly stable level throughout 80, 90, and 100% of the total duration of the free recall period (a 'baseline-shift' model). Figure 4b depicts the predictions of these models. Adding

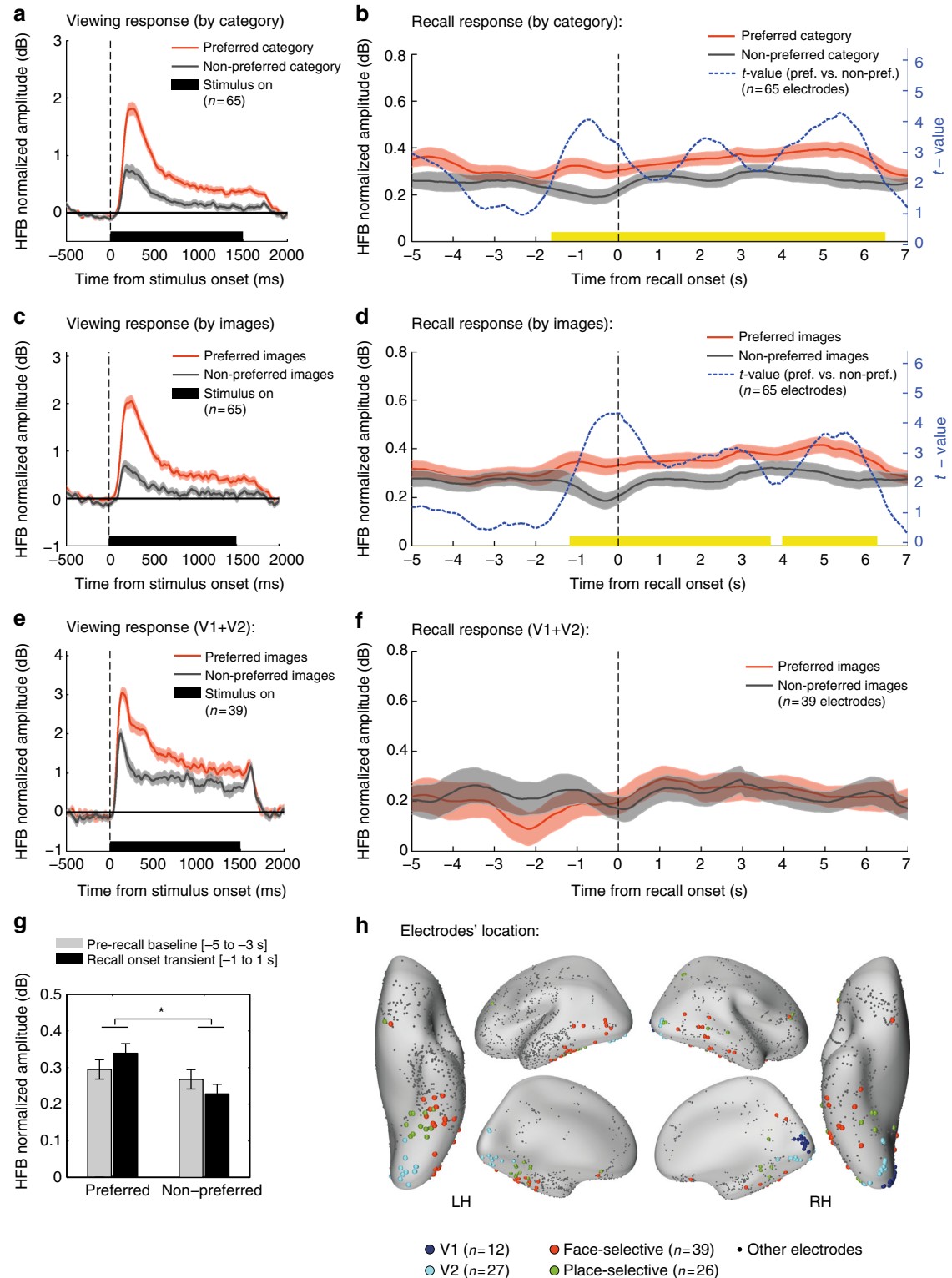

transient bursts to the signal resulted in a stronger gain preferentially in the top percentiles (reflecting a right-skewed distribution); adding bursts with a more realistic amplitude resulted in a negligible effect on the original signal distribution; finally, adding a 'baseline-shift' resulted in a similar gain across all percentiles.

In the actual experimental data, the distribution of amplitude values during recall of the preferred category showed a modulation that was more compatible with the constant "baseline-shift" model: a significant amplitude gain in all percentiles (mean gain: 0.06 dB, $p < 0.001$, preferred category minus non-preferred category, Wilcoxon signed-rank tests), with no significant inter-percentile differences ($p > 0.84$, Kruskal–Wallis test) (Fig. 4c, d).

**Transient differential signals anticipating individual recall events.** Although our results suggest that the main changes in HFB amplitude reflect a sustained process that persists throughout the recall period, it was of interest to examine the activity profile time locked to the actual recall moments, as reported verbally by the patients.

To do this, we extracted the onset of each verbal report by analyzing the voice recordings of each patient, following the previously published procedure[4] (see Methods section). We then normalized the HFB signal by the geometric mean amplitude during resting-state and smoothed it using a 1000 ms triangular window to compensate for uncertainties regarding the latencies of underlying neuronal events relative to speech onset. For each electrode, we computed the event-related HFB response time-locked to the onset of each verbal report of recall, and separately averaged responses during recall of items from the 'preferred' and the 'non-preferred' category. Finally, we computed the grand-average responses across all category-selective electrodes in a time window of −5 to 7 s relative to speech onset. Recall events that were not separated by at least 5 s from the previous recollection were excluded from the analysis.

In addition to a sustained amplitude increase during recall of items from the preferred category (as expected from the baseline-shift results described above), the recall moment was preceded by a transient differential activity: while recall of items from the preferred category involved a small amplitude increase, recall of non-preferred items involved a small amplitude decrease (see Fig. 5a, b; Fig. 5h shows electrode location). A time-point by time-point, successive, two-tailed, paired $t$-test comparing the preferred vs. the non-preferred category, indicated a quick rise in differential activity that anticipated the onset of verbal report by 1500 ms ($p < 0.05$, cluster based correction using Monte Carlo simulation with 10,000 iterations in which the 'preferred' and 'non-preferred' labels were randomly shuffled over electrodes); Fig. 5.

To increase sensitivity to recall-triggered responses and reinstatement of visual information regardless of category, we carried out a complementary analysis in which we contrasted in each electrode the responses during recall of 'preferred images' (top 10 items that preferentially activated the electrode during viewing) and 'non-preferred images' (bottom 10 items that least activated the electrode during viewing). As this contrast excludes all 'borderline' exemplars that activated the electrode in a partial manner, it should be more sensitive in detecting item-specific reactivation than the contrast of categories. Moreover, this contrast allowed comparing category selective electrodes to early visual electrodes (V1 + V2), as both electrode groups manifested a clear preference to some exemplars during viewing (not necessarily related to a specific category) (Supplementary Fig. 5).

Similar to the previous contrast, recall events that were not separated by at least 5 s from the previous recollection were excluded from the analysis. The results indicated a clear recall-triggered response in high-order category selective electrodes, involving a small amplitude increase/decrease for preferred/non-preferred images respectively (peak differential amplitude: 0.17 dB, $p < 0.05$, cluster based correction, Fig. 5c, d; see Supplementary Fig. 5 for individual electrode data). In contrast to high-order category selective visual electrodes, early visual electrodes showed no significant response ($p =$ NS, cluster based correction, Fig. 5e, f). To further test the generalization of this recall-triggered effect in category selective electrodes, we fitted the data with a mixed-effects model using the fixed factors of 'preference' (preferred/non-preferred) and 'time-bin' (baseline (−5 to −3 s)/recall onset transient (−1 to 1 s)), and the random factors of 'patient' and 'electrode'. We found a significant preference effect with higher activity levels for preferred images (F(1,192) = 13.71, $p < 0.001$), and a significant interaction between preference and time-bin (F(1,192) = 5.04, $p = 0.026$) (Fig. 5g).

**Baseline shift reduction prior to intrusion errors.** To further test the idea that the observed shifts in baseline activity reflect a top−down boundary setting mechanism, we examined the activity in category selective electrodes when an item from the non-designated category was erroneously retrieved (i.e., an extra-category intrusion). In order to perform this analysis, the normalized HFB signal in category selective electrodes was time-locked to the onset of the intrusion error. The intrusions that occurred when the electrodes' preferred category was targeted were separately analyzed from the intrusions that occurred when the non-preferred category was targeted. If the observed category-specific baseline shift indeed reflects a boundary setting process, then prior to the intrusion onset those shifts should be substantially reduced, allowing for items from the non-designated category to pop-up. The results indicated that until ~3 s prior to

**Fig. 5** HFB response time-locked to the onset of individual recall events. **a** Stimulus response in category selective electrodes to images from the preferred and non-preferred categories during the picture viewing stage. **b** Recall-related response in the same electrodes time-locked to the onset of individual recall events, showing both a sustained baseline-shift and a transient event-related component. The transient component involved both amplitude increase and decrease anticipating the recall of items from the preferred and non-preferred categories, respectively. The blue line presents $t$-values from a successive paired $t$-test comparing recall of preferred vs. non-preferred category. Significant time bins are indicated in yellow ($p < 0.05$, cluster-based correction using Monte Carlo simulation). Shaded areas reflect unbiased within electrode corrected 95% confidence interval[72]. **c, d** an alternative contrast applied to the same electrodes showing stimulus response of preferred images (top third responses) and non-preferred images (bottom third responses) during the picture viewing stage, and the corresponding responses during recall (same notations as in **b**). Here again a transient differential activity component appeared to anticipate the recall event by ~1.5 s. **e, f** Applying the same contrast (i.e. preferred vs. non-preferred images) in early visual electrodes (V1 + V2) did not reveal any significant recall-related response. **g** Comparing recall onset response in category selective electrodes to the immediate prerecall baseline using a mixed-effects model revealed a significant interaction between preference and time-bin (F(1,192) = 5.04, *$p < 0.05$), further supporting the observation that HFB amplitude was increased/decreased during recall of preferred/non-preferred images, respectively. **h** Anatomical location of face-selective (red), place-selective (green), V1 (blue) and V2 (light blue) electrodes on an inflated template brain. See Supplementary Table 2 for further details

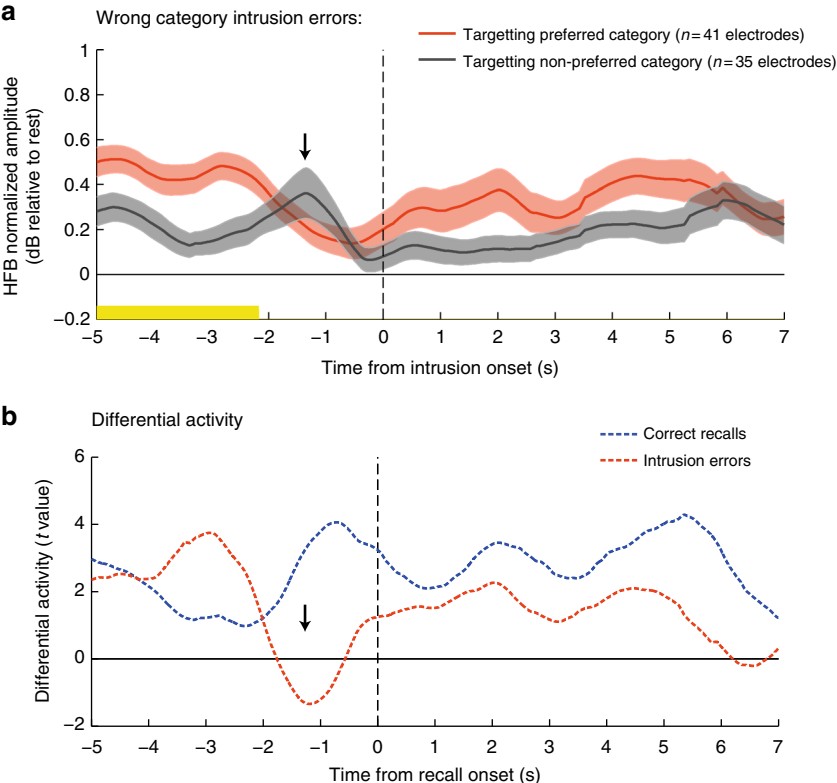

**Fig. 6** Recall errors: intrusion of items from the non-designated category. **a** HFB activity in category selective electrodes during events in which an item from the non-designated category was erroneously retrieved (i.e., extra-category intrusion). Intrusions that occurred when the preferred category was targeted are plotted in red; intrusions that occurred when the non-preferred category was targeted are plotted in gray. Note that 2 s before the onset of an intrusion error, the enhancement of the preferred (targeted-) category was transiently reduced until the opposed category has taken the lead. Shaded areas represent SEM. Significant time bins are indicated in yellow ($p < 0.05$, cluster-based correction using Monte Carlo simulation). **b** Time-course of consecutive two sample $t$-tests contrasting preferred vs. non-preferred categories during intrusion errors (blue) and during correct recollections (red; taken from Fig. 5). Note the transient decrease in $t$-values (black arrow) reflecting both baseline shift reduction and a transient reactivation of representations from the non-designated category

the intrusion's onset there was still a steady-state enhancement of the targeted category (reflected in a significantly higher HFB amplitude when targeting the electrodes' preferred category: consecutive two sample $t$-tests, $p < 0.05$, cluster based correction); However, 2 s before the occurrence of an intrusion error, the steady-state enhancement was transiently reduced until the opposed category has taken the lead, reflecting items from the non-designated category that were erroneously recalled (Fig. 6).

**Baseline shift magnitude reflects ongoing retrieval efforts**. To further examine the link between baseline shifts and ongoing retrieval efforts, we analyzed unique occasions during the free recall period, in which patients started to indicate that they were unable to recall more items. To encourage the patients, the experimenter, in these occasions, gave a standard prompt such as: 'Do you remember any other pictures of faces?'; Since these prompt events capture moments in which there was no overt recollection, and top−down search efforts were starting to decline —we predicted that there will be a parallel reduction in the magnitude of the baseline shift prior to the prompt. On the other hand, following the experimenter's prompt, we predicted that the baseline shift will recover. Critically, if the baseline shift indeed reflects a top−down mechanism for targeting a specific category, then category selective electrodes should enhance their activity following the prompt in a content-selective manner, i.e., only when the electrodes' preferred category was being prompted. To perform this analysis, HFB amplitude was normalized by the pre-

prompt baseline (−5 to −1 s). Only prompts that were separated by at least 7 s from the next recall event were included. The results indicated that following prompt offset, the category-specific baseline shift was indeed recovered (a time-point by time-point, consecutive, two-sample $t$-test comparing 'preferred' vs. 'non-preferred' prompts; $p < 0.05$, cluster based correction, $n = 65$ electrodes) (Supplementary Fig. 6a). Interestingly, examining the prompt-triggered responses in prefrontal and parietal areas (Methods section), revealed a significant activation in the left anterior ventrolateral PFC and the dorsal parietal cortex (DPC) (Supplementary Fig. 6b).

**Modulation of ultra-slow HFB fluctuations**. We next examined whether the free recall period was associated with any modulation in the power of ultra-slow HFB activity fluctuations (<1 Hz). Note the important distinction between this second-order analysis of the HFB time-course and the more direct analysis of the raw LFP power spectra, which are reported in the next subsection entitled "low-frequency power changes in the raw LFP".

Our analysis revealed that the power of ultra-slow HFB fluctuations in category selective electrodes was enhanced during recall as compared to resting-state. More specifically, normalizing the free recall spectrum by the resting-state spectrum in each electrode revealed a significant power increase that was restricted to ultra-slow frequencies between 0.09 and 0.25 Hz ($p < 0.05$, FDR corrected, Wilcoxon signed-rank test, $n = 65$ electrodes; Fig. 7a). Interestingly, this power increase was not content-

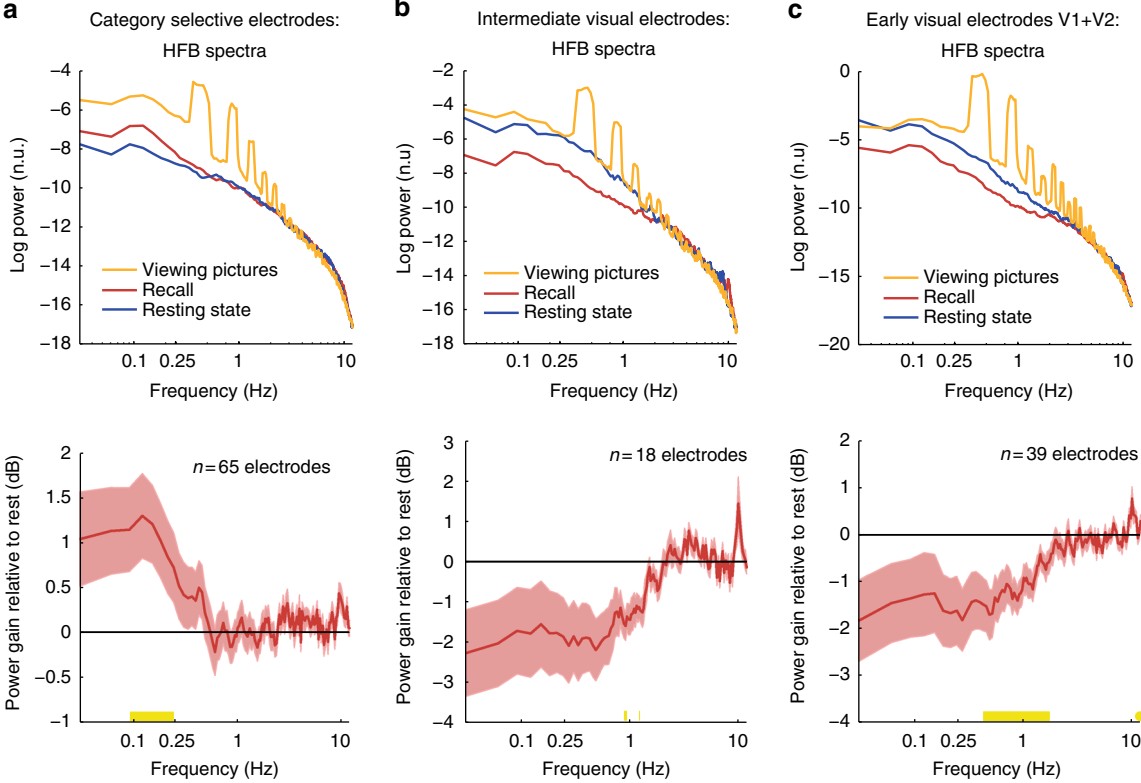

**Fig. 7** Spectral analyses of ultra-slow HFB activity fluctuations (<1 Hz). Spectral analysis of the HFB signal during the three main experimental conditions: resting-state (blue), free recall (red) and picture viewing (orange). Note the peak around 0.4 Hz and its harmonics during picture viewing, which reflects the rate of stimulus presentations. The bottom row depicts power gain relative to resting-state. **a** Category selective electrodes manifested an increase in the power of ultra-slow modulations of HFB amplitude during recall in the range of 0.09–0.25 Hz. Yellow marks on the x-axis represent the significant frequencies ($p < 0.05$, FDR corrected, signed-rank test). **b**, **c** In contrast to category-selective electrodes, early visual electrodes showed a significant reduction in the power of ultra-slow HFB fluctuation during free recall in a frequency range of 0.27–1.77 Hz; non-category-selective intermediate visual electrodes showed a similar trend with maximal power reduction between 0.88 and 1.25 Hz ($p < 0.05$, FDR corrected, signed-rank tests). Shaded areas represent SEM

specific: comparing the power of HFB fluctuations during recall of the preferred vs. the non-preferred category did not reveal significant differences (all $p$-values > 0.05 after FDR correction) (Supplementary Fig. 7). In contrast to category-selective electrodes, early visual electrodes showed a significant reduction in the power of ultra-slow HFB fluctuation during free recall in a frequency range of 0.27–1.77 Hz ($p < 0.05$, FDR corrected, $n = 39$ electrodes); non-category-selective intermediate visual electrodes showed a similar trend with maximal power reduction between 0.88 and 1.25 Hz (Fig. 7b, c).

**Low-frequency power changes in the raw LFP**. While the main focus of our study was changes in HFB amplitude, we also examined the raw ECoG signal (i.e., LFP) to determine whether other spectral changes were evident in lower frequencies (1–50 Hz) during the free recall. To that end, we compared the power spectra of the raw LFP signals during rest, picture viewing and free recall (Methods section). Within the free recall, we separately analyzed periods of overt recollections (i.e., 5-s segments centered on recall onsets), and the presumed "search" intervals between consecutive recall events (i.e., inter-event intervals (IRI): 5-s non-overlapping segments starting from recall offset until 2 s before the onset of the next recall event).

Consistent with previous observations[24], we found a widespread decrease in low-frequency power (1–9 Hz) during recall events, compared to both resting state and IRI (mean power reduction: −0.95 dB, $p < 0.05$, FDR corrected, recall vs. IRI,

signed-rank test; Fig. 8). However, these recall-related power changes were not content-selective (Supplementary Fig. 8). Intermediate visual electrodes (non-category-selective) showed a similar low-frequency power decrease during recall events (1–4 Hz; $p < 0.05$, FDR corrected, recall vs. IRI, signed-rank test), with a small power increase in the alpha band (peaking around 10.8 Hz) during both recall events and IRI ($p < 0.05$, FDR corrected, recall and IRIs vs. rest, signed-rank test). Early visual electrodes exhibited a significant power decrease between 1 and 8 Hz during recall ($p < 0.05$, FDR corrected, recall vs. IRIs, signed-rank test), and a significant power increase in the alpha band (peaking around 10.4 Hz) during both recall events and IRI ($p < 0.05$, FDR corrected, recall and IRI vs. rest, signed-rank test). Note that the increase in alpha power in those electrodes was significantly higher during recall as compared to IRI ($p < 0.05$, FDR corrected, signed-rank test). All in all, these results show that delta and theta band power across the visual hierarchy was decreased during memory search (IRI), and then further decreased during the overt recollection.

**Recall-related activity in prefrontal and parietal electrodes**. Finally, to relate our results to a large body of literature linking specific prefrontal[25–30] and parietal[21, 31, 32] regions to the strategic control of retrieval processes, we carried out the following hypothesis-based analysis. We created six regions of interest based on the literature that included the left anterior VLPFC (aVLPFC), right and left mid-posterior VLPFC (mid-VLPFC),

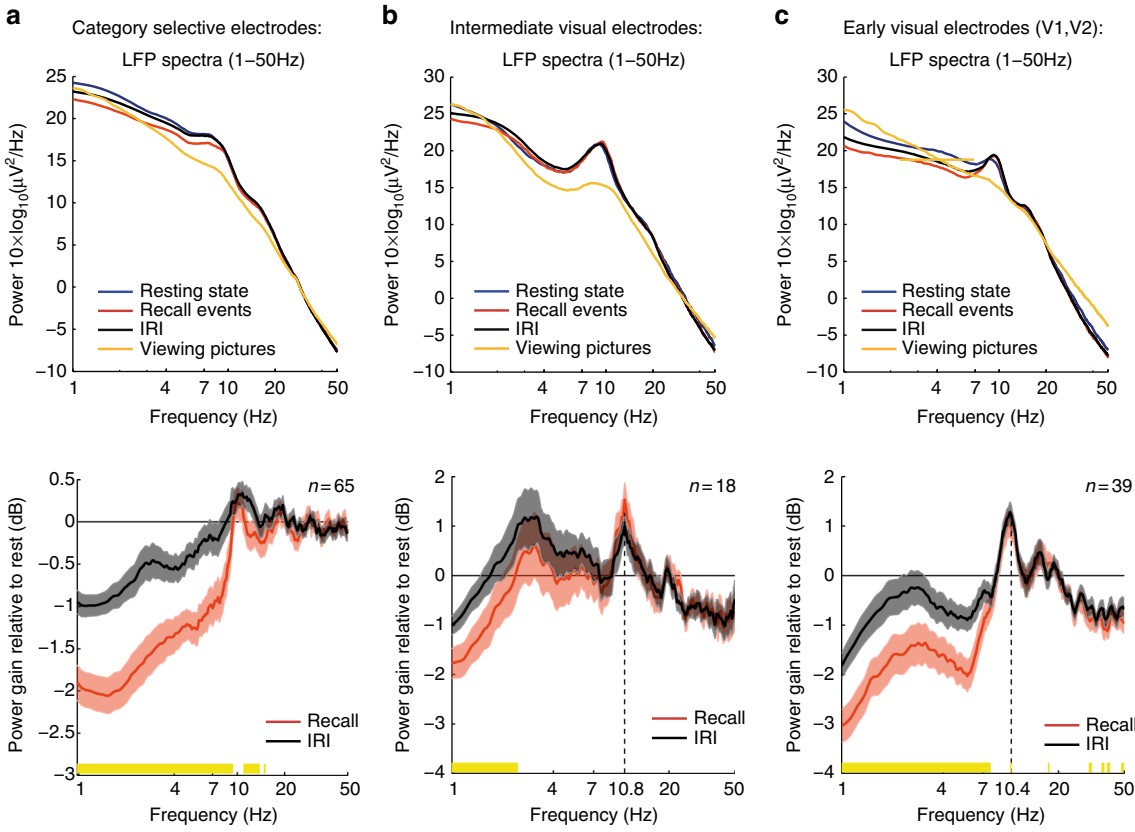

**Fig. 8** Spectral analysis of the raw LFP signal (1–50 Hz) during recall, inter-recall intervals, and picture viewing. Spectral analysis of the raw LFP signal during resting-state (blue), recall events (red), inter-recall intervals (black), and picture viewing (orange). The bottom row depicts power gain relative to resting-state. **a** Category selective electrodes: note the significant power decrease between 1–9 Hz and 11–15 Hz during recall events in comparison to IRI (significant frequencies are marked in yellow: $p < 0.05$, FDR corrected, signed-rank test). In contrast to early and intermediate visual electrodes, there was no increase in the alpha band (8–12 Hz) relative to rest. **b** Intermediate non-category-selective visual electrodes manifested a recall-related power decrease that was restricted to the delta band (1–4 Hz; significant frequencies are marked in yellow). Comparing recall events and IRI to resting state revealed a significant power increase in the alpha band (peak frequency: 10.8 Hz, $p < 0.05$, FDR corrected, signed-rank test). **c** Early visual electrodes (V1 and V2) manifested a recall-related power decrease between 1 and 8 Hz (significant frequencies are marked in yellow), and a significant power increase in the alpha band relative to rest (peak frequency: 10.4 Hz, $p < 0.05$, FDR corrected, signed-rank test). Note that power increase in the alpha band was significantly higher during recall compared to IRI ($p < 0.05$, FDR corrected, signed-rank test). Shaded areas represent SEM

right DLPFC, dorsal parietal cortex (DPC) and ventral parietal cortex (VPC) (Fig. 9a; Methods section). We then examined the overall HFB activity in these regions during the free recall period relative to resting-state. We computed the median HFB amplitude separately for overt recollection periods (2 s before speech onset until the offset of each recall event) and for the presumed memory search periods (inter-recall intervals, all complementary time points). Electrodes in VLPFC exhibited a significant increase in median HFB amplitude relative to rest during both memory search and overt recollection (mean amplitude increase: aVLPFC, 12.63%; left mid-VLPFC, 10.67%; right mid-VLPFC, 13.38%; Wilcoxon sign-rank test, $p < 0.05$, FDR corrected). Parietal electrodes showed a similar increase in median amplitude, reaching statistical significance only during periods of overt recollection (mean amplitude increase: DPC, 15.89%; VPC, 13.03%; Wilcoxon sign-rank test, $p < 0.05$) (Fig. 9b). A pairwise comparison between search and recall within each ROI revealed a significant amplitude enhancement during recall in all regions of interest except for the DLPFC ($p < 0.05$, FDR corrected, mixed-design ANOVA with fixed factors of 'retrieval-phase' (search/recall) and 'ROI', and random factors of 'patient' and 'electrode'; Methods section).

Next, we time-locked the HFB signal in each ROI to the onset of individual recall events, normalized the response by a prerecall baseline (−5 to −3 s relative to speech onset) and averaged all

responses within an electrode, excluding recall events that were not well separated from the previous recollection (<5 s separation). Finally, we computed the grand-average response across all electrodes within each ROI.

Electrodes in VLPFC showed a significant recall-triggered response (paired $t$-test against baseline, $p < 0.05$, cluster based correction; Fig. 9c), which, interestingly, followed the onset of visual reactivation (i.e., differential activity in category selective electrodes; Fig. 5b) by more than 500 ms. Parietal electrodes showed a more transient recall-triggered component that preceded the onset of visual reactivation by ~1 s (Fig. 9c). Following this transient response, VPC electrodes showed a persistent engagement throughout the verbal report.

## Discussion

Taken together, the results of this study show that there is a content-specific, rapid, and sustained increase in neuronal activity —a "baseline shift"—associated with the attempt of the patients to recall a specific category of images. Thus, our results provide a straightforward biasing mechanism for setting top–down categorical boundaries during free recall. In this respect they are compatible with previous models linking memory search to attention-like processes—the attention to memory (AToM)

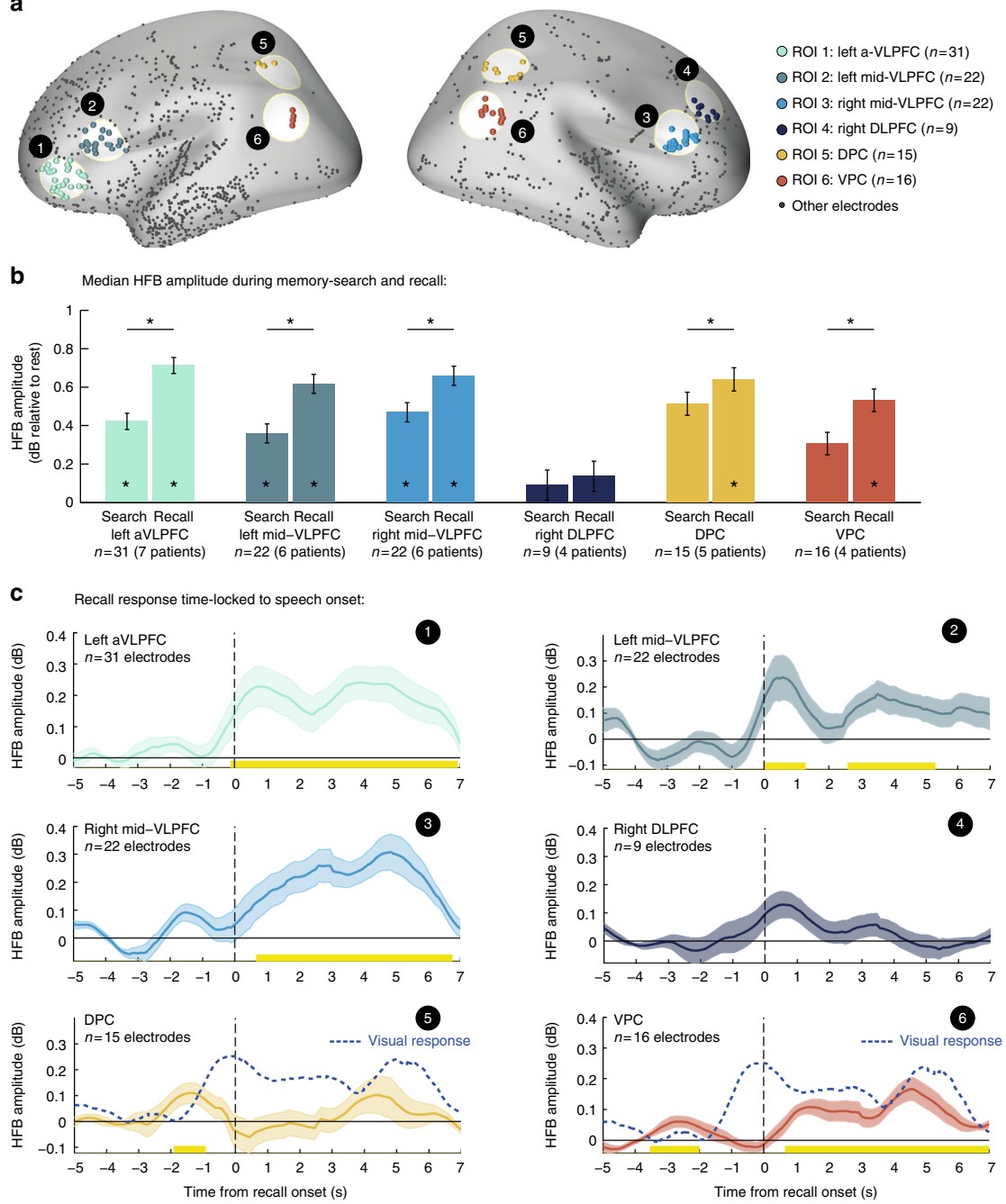

**Fig. 9** Prefrontal and parietal ROI analysis. **a** Anatomical location of each region of interest (ROI). **b** Mean median HFB amplitude during overt recollection periods and during inter-recall intervals (i.e., presumed memory search periods). Significant differences were found between search and recall in all regions of interest except for the DLPFC (*$p < 0.05$, FDR corrected). Error bars represent SEM of within electrode differences estimated by the mixed-effects model. **c** HFB response in each region time-locked to the onset of individual recall events. Note the activation latency of prefrontal regions, which followed the response of category selective electrodes (Fig. 5). Parietal regions on the other hand showed a small transient response anticipating the differential activation of category-selective visual electrodes (dashed blue line). Abbreviations: aVLPFC—anterior ventrolateral prefrontal cortex (~BA10/47); mid-VLPFC—mid-posterior ventrolateral prefrontal cortex (~BA45); DLPFC—dorsolateral prefrontal cortex (~BA8/9); DPC—dorsal parietal cortex (~BA39/7); VPC—ventral parietal cortex (~BA39/40)

model[21, 32], and provide direct empirical support to the idea of an internally maintained cue that acts to bias retrieval competition in favor of specific classes or categories, an idea which is widespread in the cognitive modeling memory literature[16–19] and plays an essential part in Tulving's notions of retrieval mode[20].

The current results are also consistent with a recent ECoG study by Morton et al.[33], demonstrating that reactivation of category-specific patterns during free recall appears most prominently in the temporal cortex and MTL, but not in early visual areas. Morton et al.[33] results also demonstrated a link between category-specific patterns in temporal regions during encoding and subsequent recall performance, thereby complementing the results reported here.

A critical issue concerning the present results is the distinction between individual item recall and the top–down boundary setting process. Taking advantage of the high signal-to-noise ratio

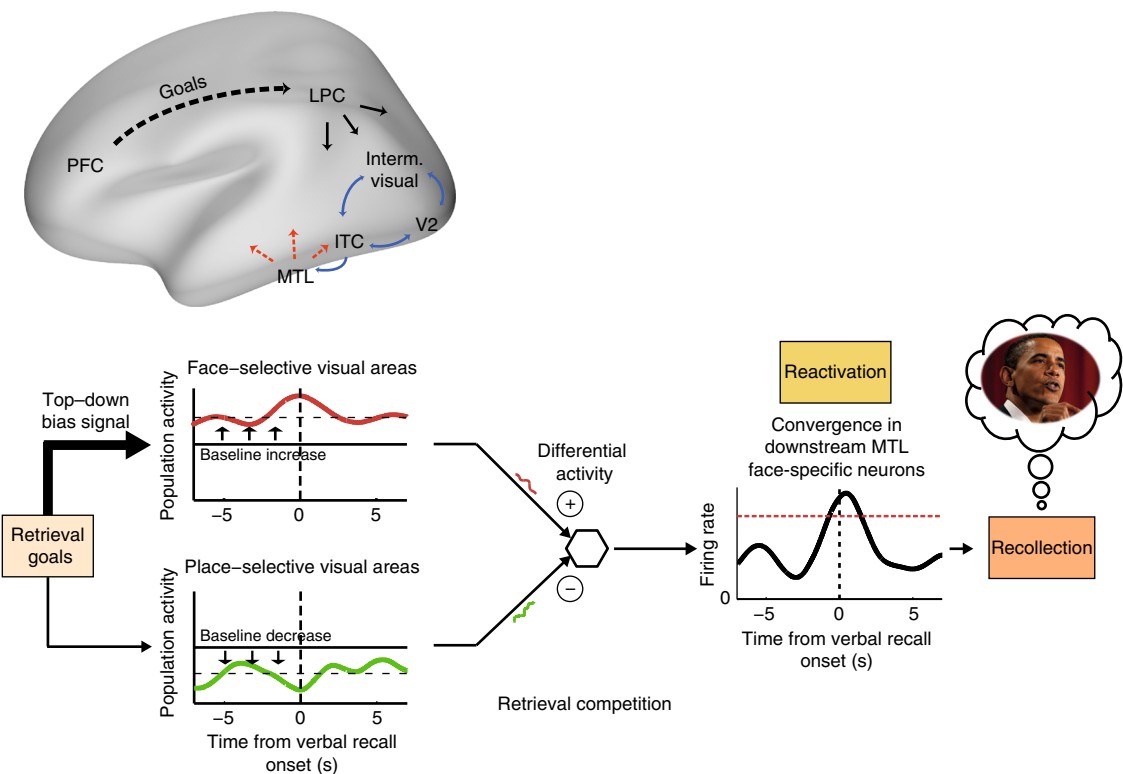

**Fig. 10** Proposed model for top−down control during free recall: targeting face-related memories. Top−down control signals are thought to originate in the prefrontal cortex and to be projected to visual areas through fronto-parietal networks. Such signals modulate the baseline activity in content-specific visual areas in accordance with retrieval goals, thereby biasing the retrieval competition towards specific memories. Stochastic fluctuations in neuronal activity within such maintained "attentional set" may occasionally reactivate the desired mnemonic representations in downstream neurons, where visual inputs converge. The overall differential activity between face-selective neural ensembles and other category-selective populations facilitates the reactivation of downstream MTL neurons, which in turn evoke the relevant recall event. An awareness threshold (red dashed line) is hypothesized to account for the long (1–2 s) delay between the onset of MTL activation and the verbal report. Abbreviations: MTL—medial temporal lobe, ITC—inferior temporal cortex, LPC—lateral parietal cortex, and PFC—prefrontal cortex

and temporal resolution of ECoG we show, surprisingly, that most of the category-selective activations during recall in high-order visual areas were associated with a goal-directed baseline-shift rather than transient reactivation of visual content during the recall event proper.

Support for this interpretation comes from the fact that amplitude increase was rather constant throughout the free recall period, even though early recall stages included significantly more recall events than later stages. Second, the baseline-shift effect remained significant even when excluding from the analysis data points with temporal proximity to recall events, suggesting that transient event-related activations did not contribute much to the overall amplitude increase during the free recall period. Third, the statistical properties of the distribution of HFB amplitude values were more consistent with a baseline-shift model rather than with the accumulation of multiple transient reactivations. And finally, category-specific activations tracked retrieval efforts, regardless of recall success.

Further support to the latter point also comes from the observation that baseline amplitudes were higher when a category was targeted second in the order of recall. This result likely reflects the increased difficulty performing the second retrieval, which was more challenging due to the larger temporal distance from the encoding phase and the greater interference from the recently retrieved (opposed-) category.

It is important to clarify that the baseline shift phenomenon observed here likely constitutes a general boundary setting mechanism during free recall. The focus on category-selectivity in

the present study was motivated by the inherent category-selectivity found in high-order visual representations, which is also conveniently spatially segregated in the cortex[23]. This robust selectivity enabled a straightforward comparison of the functional selectivity of the recorded electrodes found during viewing, with that emerging during the targeted recall.

In addition to the baseline shift phenomenon, we also observed a more transient response time-locked to individual recall events. Surprisingly, comparing HFB amplitude during recall of preferred vs. non-preferred images revealed that a substantial component of activity associated with the onset of verbal report was, in fact, a suppression of activity during recall of non-preferred items. Thus, item-specific activation associated with individual recall events was rather small in magnitude and appeared most prominently in the difference between the signal during recall of preferred and non-preferred items. Comparing this weak activations to the robust picture viewing responses indicates that reinstatement of high-order visual representations during recall is a strikingly more subtle effect compared to actual visual perception.

Interestingly, when examining the time course of this differential signal, it appeared to anticipate the actual verbal report by almost two seconds. A similar long anticipatory buildup during free recall has been observed previously in fMRI data[34] and in single-unit recordings of human hippocampus[4], as well as in other paradigms of free behavior[35, 36]—raising the intriguing possibility that slow stochastic fluctuations may play a role in generating spontaneous cognitive events[37]. Based on similar suggestions in the domain of free movements[36], a possible

interpretation of such anticipatory buildups could be that the process of free recall is driven, at least partially, by spontaneous ultra-slow fluctuations in neuronal activity[37, 38]. According to this hypothesis, when high-order visual representations send slow stochastic fluctuations to downstream MTL neurons, such fluctuations may occasionally cross the threshold of reactivating memory traces and lead to a spontaneous recall. To test the validity of this idea, we examined the amplitude of ultra-slow activity fluctuations across different visual areas during rest and free recall. Interestingly we found an enhancement in spontaneous fluctuations only in those electrodes that were related to the categories that were being retrieved. Early and intermediate non-selective visual areas showed the opposite behavior with a suppression of those fluctuations. The latter suppression may reflect an attempt of the system to reduce stochastic activity from representations that are irrelevant to the recall process. These results line up nicely with the suggested hypothesis and provide further support to the involvement of slow fluctuations in the generation of spontaneous cognitive events.

Investigating the recall process from a more global, whole brain perspective revealed a continuous HFB activity during the free recall task in VLPFC electrodes. Interestingly, an additional bilateral enhancement of VLPFC activity was observed during events of overt recollection. Critically, such enhancement appeared to begin only after the reinstatement of visual information had already begun. The latter observation may indicate that individual recall events were not directly triggered by activity in those prefrontal regions. Furthermore, the rather long delay between ITC activation and prefrontal responses highlights the gradual evolution of spontaneous recall events. This observation is compatible with previous theoretical suggestions[39, 40], positing a two stage recollection process mediated by the hippocampus—a fast subconscious stage, involving reactivation of hippocampal-neocortical memory traces, and a slower conscious one, involving cortical processes that operate on the retrieved content.

Compatible with the dominant view of VLPFC function during strategic retrieval[27, 28], our results demonstrated that electrodes in the left mid-VLPFC increased their activity a few hundred milliseconds after the onset of recall, consistent with a post-retrieval selection process. The robust, significant increase in HFB amplitude during memory search both in left anterior and right VLPFC supports the idea that these regions mediate a steady top–down control process.

Finally, consistent with the AToM model[21, 32], both DPC and VPC showed anticipatory signals during individual recall events which may attribute to transient shifts in internal attention. The persistent activity in VPC throughout the verbal report may suggest an active role in maintaining an internal representation of the retrieved material[22, 41, 42].

When taken together, how can the above results account for boundary setting during free recall? In our model, the enhanced baseline activation would bring the set of desired target representations closer to the recall threshold, hence biasing performance to the desired category. Based on previous studies[4, 40, 43], such a model necessitates positing a downstream station in the medial temporal lobe (MTL), where signals from category-selective visual areas normally converge. Maintaining an elevated baseline activity in e.g., face-selective areas during free recall would lead to an inherent bias toward faces in the ongoing input arriving at the MTL, facilitating the reactivation of neural ensembles that encode for face-related exemplars (see illustration in Fig. 10). Once a specific memory trace is reactivated, an iterative interplay between the hippocampus and the neocortex[40, 44–46] may allow the further orchestration of the reinstatement of visual details.

A continuous top–down bias is most likely maintained by a cognitive control mechanism that involves the ventrolateral prefrontal cortex[27–29, 47] and lateral parietal regions[21, 22, 32], which were shown to overlap, at least partly, with mechanisms that govern voluntary allocation of attention during perception[31, 48].

To conclude, our findings propose a novel mechanism that can account for the remarkable ease and precision by which freely recalled items remain within required boundaries. By recording intracranial ECoG signals during a categorized free recall task, we were able to show that such boundary setting involves a steady goal-directed enhancement of neuronal activity in brain regions associated with the targeted representations. Such content-specific "baseline shifts" constrain the retrieval competition within pre-defined behaviorally-relevant boundaries.

## Methods

**Participants.** Intracranial recordings were obtained from twelve patients with pharmacologically resistant epilepsy (eight females, mean age: 34.5 years, range: 22–55), at the North Shore University Hospital, NY. All patients were implanted with subdural intracranial electrodes for diagnostic purposes, as part of their evaluation for neurosurgical epilepsy treatment (see Supplementary Table 1 for demographic details and distribution of electrodes across individuals). All participants performed the task in their native language (10 English speakers, two Spanish speakers). The study was conducted according to the latest version of the Declaration of Helsinki, and all participants provided a fully informed consent according to the US National Institute of Health guidelines, as monitored by the institutional review board at the Feinstein Institute for Medical Research.

**Experimental task and stimuli.** The experiment was divided into two runs. Each run began with a closed-eyes resting state period of 200 s (30 s in the first two patients). Immediately afterward, participants were presented with 14 different pictures of famous faces and popular landmarks. Each picture repeated four times (1500 ms duration, 750 ms inter-stimulus interval) in a pseudorandom order, such that each presentation cycle contained all different pictures but order of pictures was randomized within the cycle. The same picture was never presented twice consecutively (see Fig. 1b for examples of stimuli). Participants were instructed to look carefully at the pictures and try to remember them in detail, emphasizing unique colors, unusual face expressions, perspective, and so on. Stimuli were presented via a standard LCD screen using Presentation software (Version 0.70, www.neurobs.com; picture size: 16.5° × 12.7° at ~60 cm viewing distance).

After viewing the pictures, participants put on a blindfold and began a short interference task of counting back from 150 in steps of 5 for 40 s. Two patients that were unable to wear the blindfold due to clinical reasons were asked to close their eyes instead. Upon completion, recall instructions were presented, guiding the patients to recall items from only one category at a time, starting with faces in the first run and with places in the second run, and to verbally describe each picture they recall, as soon as it comes to mind, with 2–3 prominent visual features. The instructions also emphasized reporting everything that came to mind during the free recall period. The duration of the free recall phase was 2.5 min per each category (5 min in total × two runs). In case the patients indicated that they could not remember any more items, they received a standard prompt from the experimenter (e.g., "Can you remember any more pictures?"). The order of the recalled categories was fixed across patients and counter-balanced between the two runs. A different set of pictures (7 per each category) was presented in each run.

**Voice recordings.** Verbal responses during the free recall phase were continuously recorded via a microphone attached to the patient's gown. The onset and offset of each recall event were extracted in an offline analysis, identifying the first\last soundwave amplitude change relevant to each utterance using Audacity—recording and editing software (version 2.0.6).

**Intracranial ECoG recordings.** Intracranial electrodes were arranged in subdural grids, strips, and/or depth arrays (Ad-Tech, Racine, WI, Integra, Plainsboro, NJ, and PMT Corporation, Chanhassen, MN). Recording sites in the subdural grids and strips were either 1 or 3 mm platinum disks with 4 mm / 10 mm inter-contact spacing, respectively. Recording sites in the depth electrodes were 2.5 mm platinum cylinders with 5 mm inter-contact spacing. Intracranial EEG signals were referenced to a vertex screw or a subdermal electrode, filtered electronically (analog bandpass filter with half-power boundaries at 0.1 and 200 Hz), digitized at 500 Hz and stored for offline analysis by XLTEK EMU128FS or NeuroLink IP 256 systems (Natus Medical Inc., San Carlos, CA). Stimulus-triggered electrical pulses were recorded along with the ECoG data for precise synchronization with stimulus onset. All recordings were conducted at the patients' quiet bedside.

**Electrode localization.** Prior to electrode implantation, patients underwent a T1 weighted 1 mm isometric structural MRI scan on a 3-tesla Signa HDx scanner (GE

Healthcare). After implantation, a computed tomography (CT) and a T1-weighted structural MRI scan at 1.5 Tesla were acquired to enable the precise localization of each recording site. The post-implant CT and MRI scans were skull-stripped and co-registered using FSL's BET and FLIRT algorithms[49–51], and were then co-registered to the skull-stripped pre-implant anatomical MRI scan. Concatenating these two co-registrations allowed visualization of the CT scan on top of the pre-operative MRI scan while minimizing localization error due to potential brain shift caused by surgery and implantation. Individual recoding sites were then identified visually on the co-registered CT, and were marked in each subject's pre-operative MRI native space, using BioImage Suite[52].

Pre-implant structural MRI scans were then processed using FreeSurfer 5.3[53] to segment and reconstruct the cortical surface of each patient. Recording sites were then snapped to the nearest vertex on the cortical surface. Contacts that were farther than 8 mm from the cortical surface were excluded from all further analyses. Following the previously published procedure[54], the three-dimensional mesh of each individual cortical surface was resampled and standardized using SUMA[55], allowing visualization of electrodes from different patients on a single cortical template ("fsaverage") while adhering to the contacts' location in relation to individual gyri and sulci.

**Preprocessing and data analysis.** All data analysis was performed in MATLAB (MathWorks) using EEGLAB[56], Chronux[57], and in-house code. Channels that were clinically identified as ictogenic, and channels that were corrupted during recording were excluded from the analysis. Preprocessing was done separately for each patient, beginning with the removal of 60 Hz interference as well as its harmonics using zero-phase Hamming windowed FIR band-stop filters (implemented in EEGLAB). Next, all electrodes were re-referenced to a common average (excluding corrupted channels) and inspected for electrical artifacts using a manual spectral examination and statistical analysis of voltage values distribution: electrodes with an averaged voltage value greater than 500 μV in data points above the ninety-ninth percentile were rejected. The remaining electrodes were further inspected for transient electrical artifacts, defined as time points in which at least 10% of electrodes showed voltage values >5 SDs (from the mean voltage in each electrode), or voltage gradient >30 μV/ms. Time windows of 200 ms around those outlier events were logged for exclusion in subsequent analyses. The notch filtered, re-referenced, artifacts-inspected ECoG data were then used for all subsequent analyses.

**High-frequency broadband signal.** High-frequency broadband (HFB) signal was defined in the present study as the mean normalized amplitude envelope of frequencies between 60 and 160 Hz (high-Gamma). This range of frequencies was used as the key electrophysiological marker of local neuronal activity[9–11].

HFB amplitude time series were computed through the following steps (Supplementary Fig. 2): (1) Band-pass filtering the ECoG signal in 20 Hz bands (i.e., 60–80, 80–100, 100–120, 120–140, 140–160) using zero-phase Hamming windowed FIR filters (order of 138) implemented in EEGLAB; (2) extracting the envelope of each narrow band signal by taking the absolute value of the analytic signal obtained from a Hilbert transform; (3) dividing each amplitude time series by its own mean (i.e., amplitude normalization); (4) averaging all normalized narrow band envelopes; and finally, (5) multiplying the averaged time series by the mean amplitude across all bands, bringing the signal back into volts. The above normalization procedure aims to correct for the 1/f decay in the EEG power spectrum that undesirably gives dominance to lower-frequency components. The above processing steps resulted in a single broadband amplitude time series that represents the mean neuronal activity recorded in each electrode contact[54, 58].

**HFB normalization.** Prior to any data analysis a preliminary normalization step was required to account for differences in the overall HFB amplitude level in different electrodes. During the viewing stage, HFB signal in each electrode was epoched relative to stimulus onset (−500 to 1750 ms post-stimulus), and re-expressed as fractional change by dividing each data point by the mean amplitude during a pre-stimulus baseline (−400 to −100 ms), averaged across all trials in the corresponding run. For the free recall period, the HFB signal was normalized by the geometric mean amplitude during resting state recorded at the beginning of each run. Since HFB amplitude, like other measures of population firing rates, tends to follow a log-normal distribution[59], prior to any statistical testing amplitude ratios were made normal by transformation to decibel (10 × log₁₀).

**Defining visually responsive electrodes.** In order to detect visually responsive electrodes we compared the post-stimulus normalized HFB responses (averaged over a time window of 100–500 ms) to the pre-stimulus baseline in each trial (−400 to −100 ms) using a two-tailed, paired t-test. As this test was done repeatedly for each electrode, all p-values (across electrodes and patients) were pulled together and the false discovery rate (FDR) was controlled ($\alpha = 0.01$)[60]. Electrodes that showed a significant HFB response were regarded as visually responsive.

To further identify the exact response latency in each visually responsive electrode, we compared the HFB amplitude in each time point to the pre-stimulus baseline using a paired t-test. Response latency was defined as the first time point in

which the HFB amplitude crossed a p-value of 0.05 and remained significant for at least 50 ms (Supplementary Fig. 2)[61].

Based on single-unit studies in monkeys reporting maximal response latency of ~180 ms in early visual areas V1 and V2[62], we defined a maximal latency criterion of 180 ms for V1 and V2 electrodes (used in conjunction with the anatomical creteria discribed below).

**Grouping of visually responsive electrodes.** Visual responsive electrodes were attributed based on their response profile and anatomical location to one of the following subgroups: V1, V2, intermediate visual areas, face-selective electrodes and place-selective electrodes.

V1 and V2 electrodes were selected based on their anatomical location as well as their response latency. Brodmann areas 17 and 18 (V1 and V2) were marked on each patient's cortical surface using the Brodmann atlas provided in FreeSurfer[63]. Visually responsive electrodes within these areas that manifested response latency of less than 180 ms (as described in the previous subsection "Defining visually response electrodes") were labeled V1 or V2 electrodes correspondingly. Further, a probabilistic retinotopic atlas[64] was used to mark visual retinotopic areas beyond V1 and V2, including V3, V4, lateral occipital regions, posterior parietal regions and inferior temporal regions. For simplicity, we refer to these areas as "intermediate visual areas", and visually responsive electrodes that fell within these regions were labeled "intermediate visual electrodes".

To define face-selective and place-selective electrodes, normalized HFB responses in visually responsive electrodes were averaged over a time window of 100–500 ms post-stimulus and log-transformed to dB. Following that, responses to faces and places were compared using a two-sample t-test and corrected for multiple comparisons using FDR ($\alpha = 0.01$). Significant electrodes that were located outside of V1 and V2 were labeled either 'face-selective' or 'place-selective' correspondingly.

Visually responsive electrodes that did not fall under one of the above subgroups were labeled "other visually responsive electrodes". A detailed map of the location and classification of each electrode is presented in Fig. 1c and Supplementary Table 2.

**Definition of prefrontal and parietal ROI.** To examine the activity profile in prefrontal and parietal electrodes during the free recall period, we created six regions of interest (ROI) based on previous reviews[25, 27, 32, 40] and selected fMRI and PET studies[26, 28–31, 41]. Prefrontal ROI included the anterior ($x = -48$, $y = 33$, $z = -9$) and mid-posterior ($x = -51$, $y = 18$, $z = 6$) portions of the left ventrolateral prefrontal cortex[27, 28]; the right mid-posterior ventrolateral prefrontal cortex[28] ($x = 54$, $y = 15$, $z = 15$); and the right dorsolateral prefrontal cortex[26, 30] ($x = 37$, $y = 33$, $z = 33$). Parietal ROI were defined about the center of mass of peak activations related to retrieval success effects[21], including the left ($x = -36$, $y = -57$, $z = 36$) and right ($x = 35$, $y = -54$, $z = 38$) dorsal parietal cortex (DPC); and the left ($x = -44$, $y = -60$, $z = 23$) and right ($x = 56$, $y = -64$, $z = 20$) ventral parietal cortex (VPC). The center of each ROI was projected onto a template of an inflated cortical surface ('fsaverage') included in FreeSurfer. Electrodes that fell within 15 mm from the projected coordinates were included in the ROI. All coordinates reported above are in MNI space.

**Spectral analysis.** Spectral analysis of the raw ECoG signals was done using the multitaper method[65] implemented in Chronux 2.10[57], an open-source MATLAB toolbox. The multitaper method attempts to reduce the variance of spectral estimates by applying to the data several orthogonal windowing functions (Slepian tapers). This procedure results in a set of independent spectral estimates that upon averaging provides an estimate with reduced variance.

Mathematically, the multitapered power spectrum of a time series is defined for a given frequency as an average over $N$ trials and $K$ tapers as follows:

$$s_x(f) = \frac{1}{k} \sum_{k=1}^{K} \left| \tilde{x}_{n,k}(f) \right|^2,$$

where,

$$\tilde{x}_{n,k}(f) = \frac{1}{N} \sum_{n=1}^{N} e^{-i2\pi f t} w_k(t) x_n(t),$$

$\tilde{x}_{n,k}(f)$ is the discrete Fourier transform of the product of the measured time series $x_n(t)$ with the $k^{\text{th}}$ Slepian taper (denoted $w_k(t)$).

Continuous intracranial ECoG signal from the different experimental conditions (i.e., resting-state, picture viewing and free recall) was cut into epochs of five seconds (non-overlapping), and spectral estimates were computed using seven tapers, resulting in a frequency resolution of 1 Hz. Spectral analysis of ultra-slow HFB fluctuations was done on the normalized HFB signal, in epochs of 25 s using four tapers—resulting in a frequency resolution of 0.1 Hz. All windowed data segments were demeaned before analysis and padded with zeros to a length of the

next power of 2 ($2^{12}$ in the raw LFP segments, and $2^{14}$ in the longer HFB segments).

**Statistical analyses**. For statistical testing, parametric methods were used for normal data. Since HFB amplitudes are not normally distributed, amplitude values were made normal by transformation to decibel ($10 \times \log_{10}$) prior to any statistical testing. For non-normal data or small sample size, we used the Wilcoxon signed-rank test. All statistical tests were two-tailed, and were corrected for multiple comparisons when necessary, either by FDR correction[60] when testing across multiple electrodes, or by cluster-based correction when testing across multiple time points in the event related analysis (Monte Carlo simulations with a threshold of $p < 0.05$ and 10,000 iterations in which condition labels were randomly shuffled over electrodes). No statistical methods were used to predetermine sample sizes; however, sample sizes were similar to those generally employed in the field. Data collection and analysis were not performed blind to the conditions of the experiments and no randomization was used.

**Mixed-effects group analysis**. Mixed-effects analyses were carried out using the LME4 package[66] implemented in R[67]. HFB amplitude values (dB) were fitted with a random intercept model formulated as follows: $Y \sim X_1 \times X_2 \times X_3 + (1|\text{Patient}/\text{Electrode})$, where $X_1$, $X_2$, and so on, are the relevant fixed factors in each analysis, and the terms in parentheses are the nested random factors of 'Patient' and 'Electrode'. Main effects and interactions were tested using the afex R package[68]. Degrees of freedom were computed using the Kenward-Roger (KR) method. Post hoc comparisons were carried out using lsmeans R package[69] and were corrected for multiple comparisons using FDR[70] ($p < 0.05$). Although including random slopes when applicable is generally recommended[71], these could not be included for our dataset since they led to over-parameterization (i.e., model unidentifiability).

**Data availability**. The data and code used in this study will be made available upon reasonable request.

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

## Acknowledgements

This study was funded by the memory-related research scholarship awarded to Y.N., in memory of Dr Willem Been, and by ICORE, the CIFAR-Azrieli program of Mind, Brain and Consciousness and the Siminovich grants to R.M. We thank T. Golan for critical reading of the manuscript and for help with electrode localization, D. Goldian for trigger-box design and assembly, P. Mégevand for assistance in conducting the experiment with the first patient, and R. Amit, O. Sharon and N. Bahat for many insightful discussions.

## Author contributions

R.M. and Y.N. conceived the study and designed the experiment. Y.N. analyzed the data. R.M. supervised the analysis. E.M.Y ran the experiments. A.D.M. supervised the experiments and all aspect of data collection. M.H. and E.M.Y. contributed to electrode localization. Y.N. and R.M. wrote the original draft. A.D.M. and E.M.Y. further contributed to the writing by reviewing and editing the manuscript.

## Additional information

**Competing interests:** The authors declare no competing financial interests.

