## [Peer Review File · Nature Communications]

Reviewers' comments:

Reviewer #1 (Remarks to the Author):

Norman et al address an important, but overlooked and, consequently under-investigated, question in the cognitive neuroscience of memory: What are the mechanisms that enable us to restrict our free recall of items to a designated category without interference from other categories? Using intracranial recordings (ECoG) in patients with epilepsy, Norman et al showed that performance on such tasks is accompanied by a baseline shift in activation of neuronal populations for the designated category. These populations are localized to higher order visual areas whose neurons show category preferences. As well, recall is accompanied by a concomitant decrease in activation for the non-preferred category, This is an excellent and innovative study, both conceptually and methodologically. I especially liked the way they used modelling to test whether an intermittent burst or baseline shift model provided the best account for their data. The analyses seem appropriate. Notwithstanding the study's virtues, I do, however, have a few concerns that I would like to see addressed in a revision.

Methodology

1. In some cases, the methodology is not clear. I may have missed it, but what is the frequency range for the HFB?
2. Why were participants required to describe the faces and places in detail, instead of just naming them?
3. What is meant by 'extra category intrusions' (Line 73)? Does that mean that on roughly 10% of the trials participants actually recalled items from the non-designated category, and belies their contention that such events are rare? If that is not the case, please explain. If it is the case, what do responses look like when such errors occur?
4. In Figure 5, neural responses to recall of items from the non-preferred category for that electrode, I imagine, refers to a 'face' population neuron responding to 'places' during the place recall task, and vice versa. Is that the case? Please clarify.
5. Why did so few neurons respond with a baseline shift? I understand why category preference was absent in the early visual areas, but why were more such responses not found in anterior temporal cortex, prefrontal cortex, and parietal cortex where there are face sensitive and place sensitive responses in fMRI in humans and in monkeys?
6. I could not tell from the data whether performance was related to the extent of baseline shift or the difference in baseline shift between preferred and not preferred items. I don't think that analysis was attempted, but it would be highly relevant to answering the question with which they began their paper.

Interpretation

1. The model the authors present at the end of the paper captures their findings well. It does, however, raise the question as to what type of signals from prefrontal and parietal cortices modulates the baseline shift in higher-order visual areas. Would signals related to the anticipatory build-up be found in these areas?
2. Reference is made to models of recall that implicate the frontal and parietal cortex, but contrary to the model presented here, in those models it is the medial temporal lobe signal that activates the information in higher-order visual areas, which in turn captures the attentional mechanisms in parietal cortex. Some discussion of these differences would be

useful. It may be that both types of models are needed and activation among these areas is reiterative.

3. Moscovitch (2008) and his colleagues (2016) posited that there is a two stage recollection process mediated by the hippocampus – a fast subconscious stage and a slower conscious one. This model resembles the one the authors proposed. Could the authors comment on that.

4. It is unlikely, I think, that given the design of the experiment, that memories of items in the non-designated category would not come to mind, even if they are not reported (see point 2 in Methodology). Is there any evidence from their protocol that this occurred, and if it did, are there any ECoG data related to it?

5. Would the anticipatory build

Minor Comments

Line 238 underlying

Line 279 Delete 'on'

Line 355 make>made

Summary: An excellent paper that, nonetheless, requires some revision.

Reviewer #2 (Remarks to the Author):

A predominant theory of episodic memory retrieval posits that the reactivation of neocortical activity patterns is critical to item retrieval and recollection. To date, evidence for human neocortex reactivation for specific stimuli has been mixed. In their manuscript Norman et al. use human intracranial recordings to measure the degree of item specific reactivation in the human ventral visual pathway during category targeted free recall. Unlike previous neuroimaging work, the authors do not find any evidence for item specific reactivation, as measured by high frequency power changes preceding free recall events. However, the authors do report a category specific elevation in baseline levels of high frequency power throughout the recall period. Overall, the authors provide detailed controls to show the subtle changes in baseline activity are reliable phenomena during free recall – however the claims of a mechanistic account for top-down boundary setting appear unjustified by the data.

Major comments:

-Experimental Task: The authors employ a clear experimental paradigm, however it is unclear how the two separate runs of the task are being pooled together. i) Specifically, I assume the two runs contain different stimuli? (It's clear there were no repeats within a run). ii) Was the order sequence of face/place recall also counter balanced across subjects, e.g. place is the first category of recall in the first task run? iii) Critically, I don't see any comparison of neural data between first and second runs of the task. This seems very important to show no order effects, and allows a between run comparison that should hold for baseline elevations – since comparisons are made to pre-task baseline periods. If the authors baseline effects are reliable, this should be born out in a between run comparison –

indeed that seems to be the impetus for the task design.

-Item recall effects: To explore item activation effects, the authors lock activity to speech onset. i) While electrode preference is one dimension of comparison, it's not clear why the item-level analysis is not first compared to the average background levels of the opposing category and the pre-task rest period. ii) In comparing item effects between preferred and non-preferred stimuli, the differences appear to be very small is effect size – with the claims or activity rise or fall being over stated. iii) To establish these claims individual electrode data should be presented, showing preferred/non-preferred differences.

-Early visual controls: V1/V2 sites are used several times as control locations. However, it appears that the critical control of mean HFB baseline change (e.g. Figure 3a) is not shown for V1/V2 sites. This is important as Figure 3f suggests that both V1/V2 sites display even greater ranges of retrieval selectivity for faces and places (based on comparison of y-axis in Fig 3d vs. 3f). Figure 3a should be created also for V1/V2 electrodes.

-Within electrode consistency: Related to the comment above, Figure 3a contains group averages between categories, what is critical to see is the difference of category for each electrode (both face selective and place selective) – otherwise these mean differences can be driven by different subsets of electrodes. A line plot of each electrode going into Figure 3a should be made for both ventral selective sites, and V1/V2 controls.

-Boundary setting: The authors introduce the manuscript through reference to human performance on category specific retrieval. However, this phenomenon is very different from specific list/item recall. For example, false memory formation can be achieved easily by semantic similarity of items/lures, and indeed it's these semantic associations that bias the likelihood of related items. Importantly, most examples of this behavior include semantically related but visually diverse items, for example faces, places and other objects pertaining to ones work. From this view, I find the psychological data motivating the manuscript to be at odds with the author's data that suggests a bias to within category retrieval (which is the task manipulation). It's in this respect that claims for a neuronal mechanism the supports free recall and limits exemplar errors to be at odds.

Minor:

-Figure 8: The brain schematic has IPS incorrectly positioned on angular gyrus, with some coloring under the label. This needs to be corrected/clarified.

Reviewer #3 (Remarks to the Author):

Norman and colleagues present a study of intracranial ECoG activity in epilepsy patients performing a categorized free recall task. They examine the ECoG signal in terms of oscillatory power during an initial study period where patients study a set of famous faces and famous places, as well as two recall periods in which the patients are cued to just recall items from one of the categories. The results focus on high-frequency band (HFB) oscillatory

power, and on fluctuations of HFB during memory search. Specifically, the authors show that there are category-specific shifts in HFB during encoding, that these same shifts occur during recall, and that HFB during inter-recall intervals shows category selectivity (and this looks like a sustained response).

The results are interesting, and build upon recent intracranial studies examining neural activity during free recall (e.g., Burke et al., 2014). The paper is well written, and for the most part the analyses are well motivated (a couple of exceptions and suggestions are noted below). There is a growing interest in the neural signals elicited during memory search tasks in fMRI and ECoG, but little work addressing how specific classes of memories are targeted. As such there will likely be interest among neuroscientists and psychologists. I think the paper, as currently framed, is significant, but perhaps not highly significant.

The claim that there is a sustained category-specific high-frequency response that indicates which category is being targeted is well supported by the analyses. The cognitive claim, that this is a mechanism for setting top-down boundaries during memory search, is reasonable, but I think requires a higher bar to be set regarding whether the results support the claim. The neural signals are related to recall performance in a rough way, in that they reflect the category being targeted, but if this signal is truly determining the boundaries of memory search, it should be possible to more directly link the neural dynamics of this signal to the responses being produced.

I could imagine a few ways to do this. One possibility that could be done with the existing data would be to relate variability in the HFB response to success in targeting the category. Another possibility would be to try to experimentally manipulate the shifting baseline category signal; the model might suggest what manipulations would be effective.

A subset of the results are not novel, as they conceptually replicate a prior study that is not cited or discussed. However, this prior study did not directly examine this idea of a shifting baseline of high-frequency activity. The prior study also examined categorized free recall in intracranially implanted patients:

Morton, Kahana, et al. (2013) Category-specific neural oscillations predict recall organization during memory search. *Cerebral Cortex*, 23, 2407-2422.

Morton et al. (2013) isn't discussed but results from this paper complement the current results. Some of the analyses from Fig. 3 of the current paper replicate some of the findings from Morton et al. 2013, using two of the same categories (famous faces and places). Specifically, Morton et al. showed (in their Figure 4) that patterns of oscillatory activity across intracranial electrodes were category-selective (looking at a different ROIs including temporal lobe, MTL, hippocampus, and early visual cortex). They showed that the strength of these category-specific patterns in temporal lobe during study predicted subsequent memory (their Fig 4B). Most relevant, they showed that category-specific patterns were reactivated during free recall (strongly in temporal lobe and MTL, but not in early visual cortex). These results line up nicely with the results reported in Figure 3 of the current paper. Morton et al did not focus on HFB activity per se, but did examine category-specific activity in 6 frequency bands including high gamma (65-100 Hz), and high gamma showed

strong category-specific activity. Morton et al. didn't present a detailed breakdown by frequency for the recall analysis though.

One of the unique contributions of the current study is the characterization of category-specific baseline activity between recalls, which was not examined by Morton et al. Putting the Morton et al. paper aside, the authors do a good job relating the current work to the prior ECoG and fMRI literature, though there is room for improvement in the discussion of prior modeling work. The authors describe a neural model for interpreting their results, and cite a recent modeling paper (Polyn et al. 2009, CMR) as being relevant. That paper describes a model that uses a contextual representation to target specific memories, which certainly is consistent with the authors' proposal. However, there are a number of models that I think are more directly relevant, as they explicitly were designed to explain how people target memories in categorized free recall. I think the best one for the authors' purposes is probably the SAM model. Raaijmakers and Shiffrin (1980) (this is a chapter in *The Psychology of Learning and Motivation*, vol 14, I was able to find it online) explicitly simulates categorized free recall, and the model uses a category-specific cue which selectively increases support for items from a specific category in a recall competition. Other related modeling papers include Gronlund and Shiffrin (1986) and Becker and Lim (2003). This idea of a retrieval cue supporting a class or category of memories is widespread in the cognitive modeling memory literature. It may be the case that the idea is underdeveloped in the neuroscientific literature, though Polyn et al. (2005) had a similar story regarding category-specific BOLD signals during free recall.

The current structure of the manuscript could be tightened. The spectral analysis of fluctuations in HFB oscillations, while interesting, is not well motivated in the current manuscript. This analysis shows that during recall, the HFB oscillations themselves show periodic variability in amplitude at a very low frequency. The importance of slow fluctuations in HFB amplitude is not established in the introduction, nor is this result returned to in the discussion. I recommend either removing this analysis or developing the motivation further.

The spectral analysis of raw LFP signal also doesn't cohere well with the narrative of the manuscript. There is value added by this section, as it allows the authors to relate these oscillatory effects to other recent studies of ECoG oscillations during recall (like the Burke et al. study). I wonder whether this section could be used to open the paper, to set the stage before zooming in to the examination of HFB activity.

Reviewer #4 (Remarks to the Author):

This paper explores how people target particular categories of studied items during recall, arguing that people accomplish this goal via an upward baseline shift in activity in these category-specific regions. Previous studies that have explored this kind of memory search have used fMRI, which does not have the temporal resolution to distinguish between category-specific activity evoked by individual recall events and "search-related" category-

specific activity that occurs in between these events. The present paper addresses this issue by using ECoG instead of fMRI.

Overall, I found the results of this study to be convincing. The finding is not surprising to me (I am not sure what other mechanism could be used to focus recall on a specific category); at the same time, it's clear to me that no other paper before this one has clearly demonstrated this search-related baseline shift. I am definitely of the mind that demonstrations of things that we assume to be true (but have not yet shown to be true) are of great value, especially given how often our assumptions turn out to be wrong.

My concerns are listed below:

- 1) One of the key control analyses in this study (aimed at showing that activity is related to search vs. recall events) removes timepoints around recalls. However, it is not clear why "5 seconds around the onset" was chosen for this control as opposed to removing the entire recall window from onset to offset. Since subjects were told to describe each image "in as much visual detail as possible," it seems possible that these recalls would extend beyond 5 seconds. The authors should provide justification for choosing this time window and consider rerunning their control analysis removing all the timepoints during which the subject was verbally recalling with some buffer in the beginning and end. If the delays between recalls are long enough, they should still have sufficient data to run the analysis.
- 2) It is unclear why the authors selected 10% as the proportion of signal to include in the "burst" model simulation. How did they choose this number? Regardless of how the number was chosen, it would be informative to compare the results of this simulation using different parameters to get a sense of how sensitive it is to parameter settings. For this simulation to be believable, it is important for the bursts to have the same distributional properties as actual recall events (e.g., as shown in Figure 5C). Although the Figure 5C results were only significant after multiple-comparison-correction in the window between -1 and 1, it looks like the separation between the red and grey lines might last for many more seconds (on the order of 5 seconds or more) – what happens if each "burst" actually persists on this timescale?
- 3) It is very unclear why the authors switch to a different contrast for the transient response analysis, compared to the baseline shift analysis (the transient response analysis contrasts preferred vs. non-preferred images; the baseline shift analysis contrasts responses to the preferred vs. non-preferred category). Perhaps the "preferred" image analysis is more sensitive to recall-triggered responses and/or it facilitates comparisons across regions? Regardless, the justification should be provided. It would also be useful to see the transient response to recalls from the preferred vs. non-preferred category, so this can be compared to the baseline-shift in those electrodes for the preferred vs. non-preferred category.
- 4) Many of the statistics in this paper appear to ignore the fact that different participants contributed different numbers of electrodes (i.e., they make no distinction between electrodes from the same vs. different participants). This approach is not ideal because it does not distinguish between within-subject and between-subject variance. It would be better to run mixed-effects analyses that distinguish between these types of variance. Alternatively, the authors could run a subject-level bootstrap analysis (i.e., resample

participants with replacement) to verify that the effect is reliable at the population level. If the authors run a bootstrap, they should also try a variant where they z-score these measures within-subject – if this analysis works, the authors can conclude that the effect is present within individual subjects (as opposed to being driven, e.g., by some participants having electrodes that are place selective at encoding and retrieval, and other participants having electrodes that are face selective at encoding and retrieval).

5) Why did the authors choose to normalize by the geometric mean HFB amplitude during rest (as opposed to, say, the arithmetic mean)? Does this choice matter?

Sincerely,

Ken Norman (I sign all of my reviews; this review constitutes the entirety of my communication with the journal – I do not fill out check boxes or ratings on the journal website)

Reviewer #1:

1. In some cases, the methodology is not clear. I may have missed it, but what is the frequency range for the HFB?

- The HFB frequency range (60-160Hz) is now indicated more clearly in the first paragraph of the results [line 79]:

“The analyses of ECoG data was focused on changes in high-frequency broadband amplitude (HFB: 60–160 Hz, also known as high-gamma), which was shown to be an excellent marker of local neuronal population activity¹⁻³.”

- This information can also be found in the methods sections [line 627]:

“High-frequency broadband (HFB) signal was defined in the present study as the mean normalized amplitude envelope of frequencies between 60–160 Hz (high-Gamma). This range of frequencies was used as the key electrophysiological marker of local neuronal population activity.”

2. Why were participants required to describe the faces and places in detail, instead of just naming them?

- This is now better explained in the revised results section as follows [line 59]:

“After a short interference task (backward counting), the patients were asked to freely recall these pictures, targeting each category (faces/places) in separate blocks. We instructed the patients to not only name but also to describe each picture they recall with 2-3 prominent visual features. This was done to ensure that the patients also retrieved visual information specific to the studied items, rather than just general semantic details.”

3. What is meant by ‘extra category intrusions’ (Line 73)? Does that mean that on roughly 10% of the trials participants actually recalled items from the non-designated category, and belies their contention that such events are rare? If that is not the case, please explain. If it is the case, what do responses look like when such errors occur?

- We thank the reviewer for raising this point. ‘Extra-category intrusions’ indeed refers to cases in which the patients erroneously retrieved an item from the non-designated category. Such events are indeed quite rare but not completely absent. Previous psychological studies that directly examined this phenomenon^{4,5} have reported an intrusion rate of approximately 8-14% of all recalled items, compatible with our results. We address this point in the revised results section [line 74]:

“The mean accuracy in the task (i.e. correct recalls) was 88.49% ($\pm 2.18\%$). The remaining 11.51% were ‘extra-category intrusions’, i.e. recall-events in which the patients erroneously retrieved an item from the non-designated category. Such intrusion rate is compatible with previous reports^{4,5}”.

- Following the reviewer’s helpful suggestion (also raised by reviewer #3) we now added an additional analysis of HFB activity time-locked to the onset of such intrusion (Figure 6). We address this issue in the revised results section [line 271]:

“To further test the idea that the observed shifts in baseline activity reflect a top-down boundary setting mechanism, we examined the activity in category selective electrodes when an item from the non-designated category was erroneously retrieved (i.e. extra-category intrusion). In order to perform this analysis, the normalized HFB signal in category selective electrodes was time-locked to the onset of the intrusion error. The intrusions that occurred when the electrodes’ preferred category was targeted were separately analyzed from the intrusions that occurred when the non-preferred category was targeted. If the observed category-specific baseline shift indeed reflects a boundary setting process, then prior to the intrusion onset those shifts should be substantially reduced, allowing for items from the non-designated category to pop-up. Figure 6 depicts the result of the analysis. As can be seen, until ~3 seconds prior to the intrusion’s onset there was still a steady-state enhancement of the targeted category (reflected in greater HFB activity for the electrode’s preferred category); However, two seconds before the occurrence of an intrusion error, the steady-state enhancement was transiently reduced until the opposed category has taken the lead, reflecting items from the non-designated category that were erroneously recalled. Figure 6b depicts the time-course of consecutive two sample t-tests, contrasting preferred vs. non-preferred categories during an intrusion error (blue line) and a correct recollection (red line; taken from figure 5a). Note the transient decrease in t-values (black arrow) reflecting both baseline shift reduction and a transient reactivation of representations from the non-designated category during the intrusion.”

Figure 6. Recall errors: intrusion of items from the non-designated category

(a) HFB activity in category selective electrodes during events in which an item from the non-designated category was erroneously retrieved (i.e. extra-category intrusion). Intrusions that occurred when the preferred category was targeted are plotted in red; intrusions that occurred when the non-preferred category was targeted are plotted in gray. Note that two seconds before the onset of an intrusion error, the enhancement of the preferred (targeted-) category was transiently reduced until the opposed category has taken the lead. Shaded areas represent SEM. **(b)** Time-course of consecutive two sample t-tests contrasting preferred vs. non-preferred categories during intrusion errors (blue) and during correct recollections (red; taken from figure 5). Note the transient decrease in t-values (black arrow) reflecting both baseline shift reduction and a transient reactivation of representations from the non-designated category.

4. In Figure 5, neural responses to recall of items from the non-preferred category for that electrode, I imagine, refers to a ‘face’ population neuron responding to ‘places’ during the place recall task, and vice versa. Is that the case? Please clarify.

- This is indeed the case and we thank the reviewer for this call for clarification. We have now extended the event-related analysis to address point #24 below, and added a new figure (Fig. 5a) that directly contrasts preferred and non-preferred *categories* (rather than items). The entire analysis is now better explained in the revised results section [line 227]. For quoted text and figures, please refer to point #24 below.

5. Why did so few neurons respond with a baseline shift? I understand why category preference was absent in the early visual areas, but why were more such responses not found in anterior temporal cortex, prefrontal cortex, and parietal cortex where there are face sensitive and place sensitive responses in fMRI in humans and in monkeys?

- As can be seen in the Figure 1C, our analysis did reveal several visually responsive electrodes in temporal, frontal and parietal regions. However, only a small portion of them showed a significant category selectivity - which is in line with previous fMRI and ECoG studies⁶⁻⁸. Nevertheless, given the sparse coverage of ECoG recordings, we may have failed to record from category selective “hotspots” located in prefrontal regions.
- Regardless of visual responsiveness or category selectivity, we did carry out an additional analysis of prefrontal and parietal ROI following the reviewer’s suggestion, which is described in point #7 below.

6. I could not tell from the data whether performance was related to the extent of baseline shift or the difference in baseline shift between preferred and not preferred items. I don’t think that analysis was attempted, but it would be highly relevant to answering the question with which they began their paper.

- As we describe in point #3 above, we were able to demonstrate that shortly before the occurrence of a recall error (retrieving items from the non-designated category) baseline shifts were transiently reduced, supporting the link between baseline shifts and recall accuracy. Nevertheless, to tackle this important issue from another direction we carried out an analysis of “prompting” manipulation that examined the link between baseline shift magnitude and ongoing retrieval efforts [line 291]:

“To further examine the link between baseline shifts and ongoing retrieval efforts, we analyzed unique occasions during the free recall period, in which patients started to indicate that they were “through” and unable to recall more items. To encourage the patients, the experimenter, in these occasions, gave a standard prompt such as: ‘Do you remember any other pictures of faces?’ – since these prompt events capture moments in which there was no overt recollection, and top-down search efforts were starting to decline – we predicted that there will be a parallel reduction in the magnitude of the baseline shift prior to the prompt. On the other hand, following the experimenter’s prompt, we predicted that the baseline shift will recover. Critically, if the baseline shift indeed reflects a top-down mechanism for targeting a specific category, then category selective electrodes should enhance their activity following the prompt in a content-selective manner, i.e. only when the electrodes’ preferred category was being prompted. Supplementary Figure 6 depicts the results of

this analysis. HFB amplitude was normalized by the pre-prompt baseline (-5 to -1 sec). Only prompts that were separated by at least 7 seconds from the next recall event were included. As can be seen, following prompt offset the category-specific baseline shift was indeed recovered (a time-point by time-point, consecutive, two-sample t-test comparing preferred vs. non-preferred prompts; significant time bins are marked in yellow, $P < 0.05$, cluster based correction). Interestingly, examining the prompt-triggered responses in prefrontal and parietal areas (see Methods), revealed a significant activation in the left anterior ventrolateral PFC and the dorsal parietal cortex (DPC).”

Supplementary Figure 6. Baseline shift recovery following an experimenter's prompt.

(a) HFB response to experimenter's prompts during preferred category recall (red) and non-preferred category recall (gray). Before the prompt is given there were no substantial differences between preferred and non-preferred categories. Following the prompt the baseline shift was recovered and the two signals became significantly different, with higher activity when targeting the preferred category ($p < 0.05$, cluster-based correction using Monte Carlo simulation). (b) HFB response to experimenter's prompts in the left VLPFC and the DPC. The other regions of interest did not manifest a significant response. Shaded areas represent SEM.

7. The model the authors present at the end of the paper captures their findings well. It does, however, raise the question as to what type of signals from prefrontal and parietal cortices modulates the baseline shift in higher-order visual areas. Would signals related to the anticipatory build-up be found in these areas?

- We thank the reviewer for raising this important aspect. To address this question we carried out an ROI analysis that is described in the revised results section [line 359, Figure 9]:

“Finally, to relate our results to a large body of literature linking specific prefrontal^{9–14} and parietal^{15–18} regions to the strategic control of retrieval processes, we carried out the following hypothesis-based analysis. We created six regions of interest based on the literature that included the left anterior VLPFC, right and left mid-posterior VLPFC, right DLPFC, dorsal parietal cortex (DPC) and ventral parietal cortex (VPC) (Fig. 9a; see Methods for further details). We then examined the overall HFB activity in these regions during the free recall period relative to resting-state. We computed the median HFB amplitude separately for overt recollection periods (2 sec before speech onset until the offset of each recall event) and for the presumed memory search periods (inter-recall intervals, all complementary time points). As shown in Figure 9b, VLPFC and DPC exhibited a significant increase in median HFB amplitude relative to rest during both memory search and overt recollection (Wilcoxon sign-rank test, $P < 0.05$, FDR corrected). A two-way mixed-design ANOVA with the fixed factors of ‘retrieval-phase’ (search/recall) and ‘ROI’, and the random factors of ‘patient’ and ‘electrode’, revealed a significant effect of retrieval-phase ($F(1,101)=31.15, P < 0.001$) as well as interaction between retrieval-phase and region ($F(5,101)=2.7, P < 0.05$). A pairwise comparison between search and recall in each ROI revealed a significant amplitude increase during overt recollection in the right mid-VLPFC, the left a-VLPFC and the left mid-VLPFC ($P < 0.05$, FDR corrected). Next we time-locked the HFB signal in each ROI to the onset of individual recall events, normalized the response by the pre-recall baseline (-5 to -3 sec relative to speech onset) and averaged all responses within an electrode, excluding recall events that were not well separated from the previous recollection (<5 sec separation). Finally, we computed the grand-average response across all electrodes within each ROI. As shown in Figure 9c, the ventrolateral prefrontal regions showed a significant recall-triggered response (paired t-test against baseline, significant time bins are marked in yellow, $P < 0.05$, cluster based correction). Interestingly, those responses seemed to begin only after the differential activity in category-selective electrodes had already begun. As for the DPC, in addition to a sustained amplitude increase relative to rest shown in panel b, we found a small transient component that preceded the differential activity observed in category selective visual

electrodes (dashed blue line). VPC electrodes also exhibited an anticipatory signal and remained active during the verbal report.”

Text in the revised Methods section [line 687]:

“To examine the activity profile in prefrontal and parietal electrodes during the free recall period, we created six regions of interest (ROI) based on previous reviews^{9,11,17,19} and selected fMRI and PET studies^{10,12–15}. Prefrontal ROI included the anterior ($x=-48$, $y=33$, $z=-9$) and mid-posterior ($x=-51$, $y=18$, $z=6$) portions of the left ventrolateral prefrontal cortex¹¹ (VLPFC); the right mid-posterior ventrolateral prefrontal cortex¹² ($x=51$, $y=45$, $z=12$); and the right dorsolateral prefrontal cortex^{10,14} ($x=37$, $y=33$, $z=33$). Parietal ROI were defined about the center of mass of peak activations related to retrieval success effects¹⁶, including the left ($x=-46$, $y=-62$, $z=32$) and right ($x=47$, $y=-59$, $z=30$) dorsal parietal cortex (DPC); and the left ($x=-37$; $y=-60$, $z=44$) and right ($x=39$, $y=-55$, $z=42$) ventral parietal cortex (VPC). *Each ROI was constructed as a sphere with 15 mm radius. All coordinates are in MNI space.* “

Text in the revised Discussion [line 478]:

“Examining the HFB activity in prefrontal and parietal electrodes revealed a continuous engagement of VLPFC and DPC during the free recall task. Interestingly, an additional bilateral enhancement of VLPFC activity was observed during events of overt recollection (Fig. 9b). Critically, such enhancement appeared to begin only after the reinstatement of visual information had already begun (Fig. 9c). The latter observation may indicate that individual recall events were not directly triggered by activity in those prefrontal regions. Furthermore, the rather long delay between ITC activation and prefrontal responses highlights the gradual evolution of spontaneous recall events. This observation is compatible with previous theoretical suggestions^{19,20}, positing a two stage recollection process mediated by the hippocampus – a fast subconscious stage, involving reactivation of hippocampal-neocortical memory traces, and a slower conscious one, involving cortical processes that operate on the retrieved content.

Compatible with the dominant view of VLPFC function during strategic retrieval^{11,12,21}, our results demonstrated that electrodes in the left mid-VLPFC increased their activity 2-3 seconds after the onset of recall, consistent with a post-retrieval selection process. The robust, significant increase in HFB amplitude during memory search both in left anterior and right VLPFC supports the idea that these regions mediate a steady top-down control process.

Finally, consistent with the AToM model^{16,17}, both DPC and VPC showed anticipatory signals during individual recall events, which may reflect the capture of bottom-up attention by a new item

retrieved by the MTL. Once a new memory item has popped up, top-down attention mediated by the DPC orients to it, which further stabilizes and enhances the relevant representations.”

Figure 9. Prefrontal and parietal ROI analysis.

(a) Anatomical location of each region of interest (ROI). **(b)** Mean median HFB amplitude during overt recollection periods and during inter-recall intervals (i.e. presumed memory search periods). Significant differences were found between search and recall in left and right ventrolateral prefrontal cortex. Except for the DLPFC, all other regions showed a significant increase in median amplitude relative to rest ($*p < 0.05$, FDR corrected). Note the large increase in DPC activity during both search and recall. Error bars represent SEM of within electrode differences estimated by the mixed-effects model. **(c)** HFB response in each region time-locked to the onset of individual recall events. Note the activation latency of prefrontal regions, which followed the response of category selective electrodes (see Fig. 5). Parietal regions on the other hand showed a small transient response anticipating the differential activation of category-selective visual electrodes (dashed blue line). Abbreviations: aVLPFC – anterior ventrolateral prefrontal cortex (~BA10/47); mid-VLPFC – mid-posterior ventrolateral prefrontal cortex (~BA45); DLPFC – dorsolateral prefrontal cortex (~BA8/9); DPC – dorsal parietal cortex (~BA39/7); VPC – ventral parietal cortex (~BA39/40).

8. Reference is made to models of recall that implicate the frontal and parietal cortex, but contrary to the model presented here, in those models it is the medial temporal lobe signal that activates the information in higher-order visual areas, which in turn captures the attentional mechanisms in parietal cortex. Some discussion of these differences would be useful. It may be that both types of models are needed and activation among these areas is reiterative.

- We thank the reviewer for pointing-out this issue, which is now addressed in the revised Discussion [line 498]:

“...Maintaining an elevated baseline activity in face-selective visual areas during free recall would lead to an inherent bias toward faces in the input signal arriving at MTL neurons, giving an advantage to the sub-population of MTL neurons that encode for face-related exemplars (e.g. Barack Obama’s face).

Reactivation of MTL representations may allow the further orchestration of visual information in an iterative interplay between the hippocampus and the neocortex^{19,22–24}. Once a specific memory is reinstated, attention can be oriented to it by lateral parietal regions¹⁷, which maintains the representations active and allow further introspection and memory judgments to take place.”

9. Moscovitch (2008) and his colleagues (2016) posited that there is a two stage recollection process mediated by the hippocampus – a fast subconscious stage and a slower conscious one. This model resembles the one the authors proposed. Could the authors comment on that.

- We thank the reviewer for drawing our attention to these highly relevant papers. We addressed the two-stage process in the following paragraph in the revised Discussion [line 483]:

“...the rather long delay between ITC activation and prefrontal responses highlights the gradual evolution of spontaneous recall events. This observation is compatible with previous theoretical suggestions^{19,20}, positing a two stage recollection process mediated by the hippocampus – a fast

subconscious stage, involving reactivation of hippocampal-neocortical memory traces, and a slower conscious one, involving cortical processes that operate on the retrieved content.”

10. It is unlikely, I think, that given the design of the experiment, that memories of items in the non-designated category would not come to mind, even if they are not reported (see point 2 in Methodology). Is there any evidence from their protocol that this occurred, and if it did, are there any ECoG data related to it?

- The reviewer's makes a valid point - it is indeed possible that items belonging to the non-targeted category nevertheless came to the patients' mind but were not reported. However, it should be noted that in the experiment the patients were specifically asked to verbally report on everything that came to mind during the free-recall period. Looking at their behavior, it seems that the patients indeed followed this instruction. For example, on average there were 4.04 utterances per run in which the patients reported items that were already retrieved (event though they knew that these events did not count). Nevertheless, we cannot rule out the possibility that items from the non-designated category were retrieved but not reported. Such events, if occurred, could be analyzed only retrospectively, which was not feasible under the current design.

Reviewer #2:

11. Experimental Task: The authors employ a clear experimental paradigm, however it is unclear how the two separate runs of the task are being pooled together. i) Specifically, I assume the two runs contain different stimuli? (It's clear there were no repeats within a run). ii) Was the order sequence of face/place recall also counter balanced across subjects, e.g. place is the first category of recall in the first task run? iii) Critically, I don't see any comparison of neural data between first and second runs of the task. This seems very important to show no order effects, and allows a between run comparison that should hold for baseline elevations – since comparisons are made to pre-task baseline periods. If the authors baseline effects are reliable, this should be born out in a between run comparison – indeed that seems to be the impetus for the task design.

- We thank the reviewer for this call for clarification and further analysis. The data in figure 3 was pooled together by averaging the median amplitude during free recall of faces/places across the two runs [line 113]:

“The median amplitude was averaged across the two runs of the task”.

- The two runs indeed contained different sets of stimuli. The order of the recalled categories was fixed across patients and counter-balanced between the two runs. We now better emphasize this in the revised manuscript [line 575]:

“The order of the recalled categories was fixed across patients and counter-balanced between the two runs (starting with faces in the first run and places in the second run). A different set of pictures (7 per each category) was presented in each run.”

- Following the reviewer's comment we now examined the reliability of the baseline shift effect and tested for possible run/order effects. Specifically, we carried out the following analysis which is now fully described in Supplementary Figure 4 [Results, line 157]:

“To test the reliability of the baseline shift phenomenon across the two runs of the task, we carried out a follow-up analysis in which we extended the two-way mixed-effects model described in Figure 3a into a three-way split-plot model with fixed factors of ‘electrode group’, ‘recalled category’ and ‘run’, and random factors of ‘patient’ and ‘electrode’. To allow a more adequate comparison between runs, prior to the ANOVA the HFB amplitude in each electrode was normalized by the mean amplitude across the entire free recall session in each run. This alternative normalization procedure was aimed at minimizing variability related to resting state differences between runs and electrodes. We found again the same significant interaction between electrode group and recalled category ($F(1,189)=24.94$, $P<0.0001$), with neither run effect nor interaction between run, electrode group and recalled-category ($F(1,189)=0.11$, $P=0.74$). We did find however, a significant interaction between run and recalled category ($F(1,189)=11.1$, $P=0.001$), which was expected given that the order of recalled categories was counter balanced between the two runs (i.e. targeting faces before places in the first run, and place before faces in the second run). Such interaction pointed to a possible recall-order effect, that is, the effect of being targeted first or second. Therefore, we carried out an additional mixed-design split-plot ANOVA, this time using the fixed factors of ‘electrode group’, ‘recalled category’ and ‘recall order’ (and the same random factors). Here again the interaction between recalled category and electrode group was highly significant ($F(1,189)=24.94$, $P<0.0001$). However, we also found a significant recall order effect ($F(1,189)=11.1$, $P=0.001$) with no further interaction (Supplementary Fig. 4a). A direct comparison between the two levels of recall-order (targeted first vs. targeted second) revealed an overall increase in HFB activity for both preferred and non-preferred categories when targeted second (that is, after the opposed category was already retrieved) ($t(189)=-3.33$, $P=0.001$).”

a Category Selective Electrodes - Order Effect:

b Baseline shift across individual electrodes (category selective):

Supplementary Figure 4. Mixed-design split-plot ANOVA for testing recall order effects.

(a) A mixed-design split-plot ANOVA using the fixed factors of 'electrode group', 'recalled category' and 'recall order', and the random factors of 'patient' and 'electrode' revealed in addition to an interaction between recalled category and electrode group, a significant recall order effect ($F(1,189)=11.10, P=0.001$). A direct comparison between the two levels of recall-order (targeted first vs. targeted second) revealed an overall increase in HFB activity for both preferred and non-preferred categories when they were targeted second, that is, after the opposed category was already retrieved ($t(189)=-3.33, p<0.01$). This result is likely to reflect an increase in cognitive demands linked to the second retrieval, which is more challenging due to the temporal distance from the encoding phase and the top-down control efforts needed in order to ignore recently-retrieved items from the opposed category. (b) A spaghetti plot showing within electrode amplitude differences between preferred category and non-preferred category retrieval.

- We discuss this finding in line 430:

"An interesting issue concerns the possibility of an ordering effect, i.e. changes in HFB activity levels depending on whether the targeted category was first/second in the order of recall. Our analysis indeed revealed that such order effects were present (Supplementary Fig. 4). This order effect is likely to reflect an increase in cognitive demands linked to the second retrieval, which was more challenging due to the larger temporal distance from the encoding phase and the top-down control efforts needed in order to suppress the more recently-retrieved items from the opposed category."

12. Item recall effects: To explore item activation effects, the authors lock activity to speech onset. i) While electrode preference is one dimension of comparison, it's not clear why the item-level analysis is not first compared to the average background levels of the opposing category and the pre-task rest period. ii) In comparing item effects between preferred and non-preferred stimuli, the differences appear to be very small in effect size – with the claims of activity rise or fall being overstated. iii) To establish these claims individual electrode data should be presented, showing preferred/non-preferred differences.
- Following the reviewer's suggestion we explored the item-activation effect both in comparison to the background levels of the opposing category and in comparison to pre-task resting-state:

As shown in the figure, there were no substantial differences between the two normalization methods, except for a greater difference in the overall baseline activity when the signal was normalized by the

baseline level of the opposing category. This difference actually reflects the same baseline shift that was reported in Figure 3a.

- To further address the reviewer’s concerns, we present in Figure 5g (see point #4 above) the results of a mixed-design ANOVA aimed at simultaneously contrasting both dimensions – electrode preference and time. In the analysis we binned the preferred/non-preferred recall responses in two time bins: immediate pre-recall baseline (-5 to -3 sec relative to speech onset) and recall onset transient (-1 to 1 sec). We then fitted the data with a mixed-effects model using the fixed factors of ‘preference’ (preferred/non-preferred image) and ‘time-bin’, and the random factors of ‘patient’ and ‘electrode’, and found a significant preference effect with higher activity levels for preferred images ($F(1,192)=13.71, P<0.001$), and a significant interaction between preference and time-bin ($F(1,192), P=0.03$), which further supports our claim about amplitude increase/decrease during preferred/non-preferred item recall.
- Since the effect size was indeed rather small, we point it out in the discussion as follows [line 451]:
“...Thus item-specific activation associated with specific recalls was rather small in magnitude and appeared most prominently in the difference between the signal during recall of preferred and non-preferred items (Fig. 5g and 5b, blue line). It should be noted that the same category selective electrodes manifested striking activations during picture viewing (Fig. 5a,c); this latter fact further indicates that reinstatement of high-order visual representations during recall is a much subtler process than actual visual stimulation.”
- Following the reviewer’s request, we present in supplementary figure 5 individual electrode data of preferred/non-preferred differences across category selective electrodes, and we refer to it in line 262.

13. Early visual controls: V1/V2 sites are used several times as control locations. However, it appears that the critical control of mean HFB baseline change (e.g. Figure 3a) is not shown for V1/V2 sites. This is important as Figure 3f suggests that both V1/V2 sites display even greater ranges of retrieval selectivity for faces and places (based on comparison of y-axis in Fig 3d vs. 3f). Figure 3a should be created also for V1/V2 electrodes.

- We addressed the reviewer’s request in Supplementary Figure 3, and refer to this analysis in the results [line 125; line 154]:

Supplementary Figure 3. HFB activity in areas V1 and V2 during the free recall period.

(a) During the free recall period electrodes in V2 showed a significant increase in median HFB amplitude relative to resting-state (** $P=0.0017$, Wilcoxon signed rank test). V1 electrode did not show this effect ($P=0.46$, Wilcoxon signed rank test). Directly comparing the two groups of electrodes using a Wilcoxon rank sum test revealed a significant difference (** $P<0.001$). (b-c) Early visual electrodes manifested a significant difference in activity levels during the free recall period between the two categories (Figure 3f). To check if those baseline modulations were linked to the category preference of these electrodes during viewing, we split V1 and V2 electrodes into two groups based on their category preference during viewing (face/place preference), and compared the median amplitudes during faces/places retrieval across the two groups in a two-way mixed-effects ANOVA (similar to the analysis presented in figure 3a). We did not find any relation between category preference during viewing and baseline modulation during free recall in these electrode (i.e. no interaction between 'category-preference' and 'recalled-category': $F(1,37)= 0.26$, $P<0.6$). Regardless of category preference during viewing, we found an intriguing overall increase in HFB amplitude during recall of faces compared to places ($F(1,37)=22.76$, $P<0.0001$).

14. Within electrode consistency: Related to the comment above, Figure 3a contains group averages between categories, what is critical to see is the difference of category for each electrode (both face selective and place selective) – otherwise these mean differences can be driven by different subsets of electrodes. A line plot of each electrode going into Figure 3a should be made for both ventral selective sites, and V1/V2 controls.

- We addressed the reviewer's request in Supplementary Figure 3c and Supplementary Figure 4b.

Revised result section [line 152; line 130]:

“...Conversely, attempting to search for a link between category-selectivity and baseline-shift in early visual cortex failed to show any significant effect (see Fig. 3f and Supplementary Fig. 3b,c) – perhaps related to the very weak category selectivity of electrodes in these areas during picture viewing.”

“...see Supplementary Fig. 4b for a spaghetti plot”

Supplementary Figure 4b:

15. Boundary setting: The authors introduce the manuscript through reference to human performance on category specific retrieval. However, this phenomenon is very different from specific list/item recall. For example, false memory formation can be achieved easily by semantic similarity of items/lures, and indeed it's these semantic associations that bias the likelihood of related items. Importantly, most examples of this behavior include semantically related but visually diverse items, for example faces, places and other objects pertaining to ones work. From this view, I find the psychological data motivating the manuscript to be at odds with the author's data that suggests a bias to within category retrieval (which is the task manipulation). It's in this respect that claims for a neuronal mechanism the supports free recall and limits exemplar errors to be at odds.

- We thank the reviewer for raising this issue. We address this comment in the discussion [line 423]:

“It should be emphasized, however, that the category-specific baseline shifts we found in visual areas are not necessarily the boundary setting mechanism for all types of targeted recall. Our choice of demonstrating the baseline shift phenomena using visual categories as the targeted boundaries was motivated by the inherent category-selectivity found in high order visual representations, which is also conveniently spatially segregated in the cortex⁷. This robust selectivity enabled a straightforward comparison of the functional selectivity of the recorded electrodes found during viewing, with that emerging during the targeted recall.”

- We agree that semantic association plays a central role in free recall, as the reviewer clearly pointed out. However, in the current study we employed an experimental paradigm that specifically targeted and enhanced visually-related categorical associations rather than other types of semantic associations. Our experimental design allowed us to track the process of boundary setting on well-characterized cortical representations – Faces and Places – that reside in separate cortical regions^{6,7}. Thus the choice of category was mandated by the methodological need to work with representations that could be monitored easily using ECoG, revealing the process of targeted memory search.

- It is important to note that despite the semantic associations between the pictures used in our experiment (Obama - White House), patients were able to target the pictures based strictly on their visual category, suggesting that the targeting of memories based on their category relations is indeed possible and effective.

16. Minor: Figure 8 - The brain schematic has IPS incorrectly positioned on angular gyrus, with some coloring under the label. This needs to be corrected/clarified.

- We have corrected the relevant figure.

Reviewer #3:

17. The claim that there is a sustained category-specific high-frequency response that indicates which category is being targeted is well supported by the analyses. The cognitive claim, that this is a mechanism for setting top-down boundaries during memory search, is reasonable, but I think requires a higher bar to be set regarding whether the results support the claim. The neural signals are related to recall performance in a rough way, in that they reflect the category being targeted, but if this signal is truly determining the boundaries of memory search, it should be possible to more directly link the neural dynamics of this signal to the responses being produced.

I could imagine a few ways to do this. One possibility that could be done with the existing data would be to relate variability in the HFB response to success in targeting the category. Another possibility would be to try to experimentally manipulate the shifting baseline category signal; the model might suggest what manipulations would be effective.

- The reviewer makes a valid point that was also mentioned by the first reviewer. We approached this issue by carrying out two analyses:

(1) Following the reviewers' suggestion, we examined the boundary setting process around extra-category intrusion errors (i.e. recall-events in which the patients erroneously retrieved items from the non-designated category). The results of this analysis are depicted in point #3 above (Figure 6).

(2) Approaching this issue from another direction – we examined the link between baseline shift magnitude and ongoing retrieval efforts by analyzing the unique occasions in which the patients indicated that they were “through” and unable to recall more items. To encourage the patients, the experimenter gave a standard prompt such as: ‘Do you remember any other pictures of faces?’ – since these prompt events capture moments in which there was no overt recollection, and top-down search efforts were starting to decline – we predicted that there will be a similar reduction in baseline shift magnitude. On the other hand,

following the experimenter's prompt, we predicted that baseline shift will slowly recover. The results of this analysis are depicted in Point #6 above (Supplementary Figure 6).

18. A subset of the results are not novel, as they conceptually replicate a prior study that is not cited or discussed. However, this prior study did not directly examine this idea of a shifting baseline of high-frequency activity. The prior study also examined categorized free recall in intracranially implanted patients (Morton, Kahana, et al., 2013). Morton et al. (2013) isn't discussed but results from this paper complement the current results. Some of the analyses from Fig. 3 of the current paper replicate some of the findings from Morton et al. 2013, using two of the same categories (famous faces and places). Specifically, Morton et al. showed (in their Figure 4) that patterns of oscillatory activity across intracranial electrodes were category-selective (looking at different ROIs including temporal lobe, MTL, hippocampus, and early visual cortex). They showed that the strength of these category-specific patterns in temporal lobe during study predicted subsequent memory (their Fig 4B). Most relevant, they showed that category-specific patterns were reactivated during free recall (strongly in temporal lobe and MTL, but not in early visual cortex). These results line up nicely with the results reported in Figure 3 of the current paper. Morton et al did not focus on HFB activity per se, but did examine category-specific activity in 6 frequency bands including high gamma (65-100 Hz), and high gamma showed strong category-specific activity. Morton et al. didn't present a detailed breakdown by frequency for the recall analysis though.

- We thank the reviewer for drawing our attention to this highly relevant article that we have missed in our literature review. We incorporated a paragraph in the revised manuscript that discusses the relation between our results and the results of Morton et al. (2013) [Line 400]:

“The current results are also consistent with a recent ECoG study by Morton and colleagues²⁵, demonstrating that reactivation of category-specific patterns during free recall appears most prominently in the temporal cortex and MTL, but not in early visual areas. Morton et. al's results also demonstrated a link between category-specific patterns in temporal regions during encoding and subsequent recall performance, thereby complementing the results reported here.”

19. One of the unique contributions of the current study is the characterization of category-specific baseline activity between recalls, which was not examined by Morton et al. Putting the Morton et al. paper aside, the authors do a good job relating the current work to the prior ECoG and fMRI literature, though there is room for improvement in the discussion of prior modeling work. The authors describe a neural model for interpreting their results, and cite a recent modeling paper

(Polyn et al. 2009, CMR) as being relevant. That paper describes a model that uses a contextual representation to target specific memories, which certainly is consistent with the authors' proposal. However, there are a number of models that I think are more directly relevant, as they explicitly were designed to explain how people target memories in categorized free recall. I think the best one for the authors' purposes is probably the SAM model. Raaijmakers and Shiffrin (1980) (this is a chapter in *The Psychology of Learning and Motivation*, vol 14, I was able to find it online) explicitly simulates categorized free recall, and the model uses a category-specific cue which selectively increases support for items from a specific category in a recall competition. Other related modeling papers include Gronlund and Shiffrin (1986) and Becker and Lim (2003). This idea of a retrieval cue supporting a class or category of memories is widespread in the cognitive modeling memory literature. It may be the case that the idea is underdeveloped in the neuroscientific literature, though Polyn et al. (2005) had a similar story regarding category-specific BOLD signals during free recall.

- Here again we thank the reviewer for drawing our attention to these highly relevant modeling papers. We now refer to them in the revised discussion [line 394].

“Our results provide a straightforward biasing mechanism for setting top-down categorical boundaries during free recall. In this respect they are compatible with previous models linking memory search to attention-like processes – the attention to memory (AToM) model^{16,17}, and provide a direct empirical support to the idea of an internally maintained cue that acts to bias retrieval competition in favor of specific classes or categories, an idea which is widespread in the cognitive modeling memory literature^{26–29}.”

20. The current structure of the manuscript could be tightened. The spectral analysis of fluctuations in HFB oscillations, while interesting, is not well motivated in the current manuscript. This analysis shows that during recall, the HFB oscillations themselves show periodic variability in amplitude at a very low frequency. The importance of slow fluctuations in HFB amplitude is not established in the introduction, nor is this result returned to in the discussion. I recommend either removing this analysis or developing the motivation further.

- The significance of this finding is now explained better in the revised discussion [line 458].

“Interestingly, when examining the time course of this differential signal, it appeared to anticipate the actual verbal report by almost 2 seconds (see Fig. 5). A similar long anticipatory buildup during free recall has been observed previously in single-unit recordings of human hippocampus³⁰, as well as

in other free behavioral paradigms^{31,32} – raising the intriguing possibility that slow stochastic fluctuations may play a role in generating spontaneous cognitive events³³. Based on similar suggestions in the domain of free movements³², a possible interpretation of such an anticipatory buildup could be that the process of free recall is driven, at least partially, by spontaneous ultra-slow fluctuations in neuronal activity (similar to the ubiquitous "resting state" fluctuations^{33,34}). According to this hypothesis, when high-order visual representations send slow stochastic fluctuations to downstream MTL neurons, such fluctuations may occasionally cross the threshold of reactivating memory traces and thereby lead to a spontaneous recall. To test the validity of this idea we examined the amplitude of ultra-slow activity fluctuations across different visual areas during rest and free recall. Interestingly we found an enhancement in spontaneous fluctuations only in those electrodes that were related to the categories that were being retrieved. Early and intermediate non-selective visual areas showed the opposite behavior with a suppression of those fluctuations (Fig. 7c). The latter suppression may reflect an attempt of the system to reduce “noise” arising from irrelevant representations. These results line up nicely with the suggested hypothesis and provide further support to the involvement of slow fluctuations in the generation of spontaneous cognitive events.”

21. The spectral analysis of raw LFP signal also doesn't cohere well with the narrative of the manuscript. There is value added by this section, as it allows the authors to relate these oscillatory effects to other recent studies of ECoG oscillations during recall (like the Burke et al. study). I wonder whether this section could be used to open the paper, to set the stage before zooming in to the examination of HFB activity.

- We thank the reviewer for pointing out this suggestion. Based on previous studies that demonstrated a tight link between neuronal firing rates and HFB (high-gamma) amplitude^{1,3} – we consider the HFB signal to be the most informative ECoG marker for local population activity. We therefore prefer to open the paper with the HFB signal and give it the main focus.

Reviewer #4

22. One of the key control analyses in this study (aimed at showing that activity is related to search vs. recall events) removes time points around recalls. However, it is not clear why “5 seconds around the onset” was chosen for this control as opposed to removing the entire recall window from onset to offset. Since subjects were told to describe each image “in as much visual detail as possible,” it seems possible that these recalls would extend beyond 5 seconds. The authors should provide justification for choosing this time window and consider rerunning their control analysis removing all the time points during which the subject was verbally recalling with some buffer in the

beginning and end. If the delays between recalls are long enough, they should still have sufficient data to run the analysis.

- We followed the reviewer suggestion and extracted from the voice recording the offset of each individual recall. The median duration of individual recall events from onset to offset was 6.05 seconds. Correspondingly, we revised the control analysis as follows [line 184]:

“First, based on the patients’ voice recordings (see Methods) we extracted all periods of overt recollection (from speech onset to offset) and recomputed the median HFB amplitude while excluding data points from 2 sec before the onset until the offset of each individual recall. A mixed-design ANOVA, identical to the one reported in Figure 3a, indicated again a highly significant interaction effect ($F(1,63)=7.91$, $P=0.007$), only this time the effect was entirely due to activity during the inter-recall intervals, when patients were involved in memory search rather than in an overt recollection.”

23. It is unclear why the authors selected 10% as the proportion of signal to include in the “burst” model simulation. How did they choose this number? Regardless of how the number was chosen, it would be informative to compare the results of this simulation using different parameters to get a sense of how sensitive it is to parameter settings. For this simulation to be believable, it is important for the bursts to have the same distributional properties as actual recall events (e.g., as shown in Figure 5C). Although the Figure 5C results were only significant after multiple-comparison-correction in the window between -1 and 1, it looks like the separation between the red and grey lines might last for many more seconds (on the order of 5 seconds or more) – what happens if each “burst” actually persists on this timescale?

- We thank the reviewer for drawing our attention to this point. First, we should clarify that the event related response indeed combined both a transient and a more sustained component. The latter in fact reflects the same baseline shift described in Figure 3. On top of this steady-state modulation there was a transient, event-related component that anticipated speech onset during the spontaneous recall events. We directly address this point in line 237 in the results sections.
- Following the reviewer’s suggestions we adjusted the ‘burst’ model to fit the actual frequency of verbal recall events (median event duration = 6.05 sec, 12.8 events per run, a total of ~25% of the entire free recall period). As shown in the revised Figure 4, we iteratively simulated signals using different sets of parameters – burst amplitude of either 2dB reflecting a clear burst dynamics, or 0.17dB reflecting a more realistic burst amplitude based on the event related analysis (peak differential amplitude between

-1 to 1 sec, Figure 5b); And bursts frequency increased incrementally from 10% to 30% of the total free recall period.

- The revised analysis is presented in the revised results section as follows [line 191]:

“...we looked at the statistical distribution of momentary HFB amplitude levels during the free recall period. A situation where the activity was dominated by many separate individual recall events should lead to a very different distribution of amplitude values compared to the situation where there was a stable overall baseline-shift.

Figure 4 depicts the results of this analysis. We first considered in a simple model how the distribution of HFB amplitude values would look if we assume no baseline-shift so that the entire response dynamics consisted of several independent bursts of activations linked to individual recall events (a "burst" model). We explored two versions of this model. The first was aimed to illustrate a clear case of sporadic burst dynamics. It was constructed by adding sporadic bursts of 2dB (convolved with a 3 sec Gaussian window) to a typical HFB signal taken from a representative face-selective electrode during recall of places. The second simulation was constructed by adding to the same representative signal sporadic bursts of 0.17dB, to simulate more realistic burst amplitude, given the observed response to individual recall events presented below (peak amplitude difference between preferred and non-preferred images, Figure 5b). Bursts' frequency was explored at 10%, 20% and 30% of the total duration of the free recall period.

In contrast to the above model we also considered the alternative case in which the activation change observed during free recall was solely due to a constant overall change in baseline activity (i.e. a constant gain of 0.1 dB) that persisted at a fairly stable level throughout 80%, 90% and 100% of the total duration of the free recall period ('baseline-shift' model). Panels b depicts the predictions of these models: Adding transient bursts to the signal resulted in a stronger gain preferentially in the top percentiles (reflecting a right-skewed distribution); Adding bursts with a more realistic amplitude resulted in a negligible effect on the original signal distribution; And finally, adding a 'baseline-shift' resulted in a similar gain across all percentiles.

The behavior of the actual experimental data of category selective electrodes during the free recall period is depicted in figure 4c-d. The distribution of amplitude values during recall of the preferred-category showed a modulation that was more compatible with the constant "baseline-shift" model: a significant amplitude gain in all percentiles ($P < 0.001$, preferred category minus non-preferred category, Wilcoxon signed-rank tests), with no significant inter-percentile differences ($p > 0.84$, Kruskal-Wallis test).”

Figure 4. Amplitude increase during retrieval reflects baseline-shift rather than transient bursts.

(a) Top: an example log-normal distribution of HFB amplitude values in a representative face-selective electrode during retrieval of the non-preferred category (black). Magenta lines illustrate distributions of simulated signals (shown below) that represent the three main alternatives for the underlying process: 'burst' model – in which we artificially inserted sporadic bursts of 2 dB (convolved with a 3 sec Gaussian window) to 10% (light-blue) 20% (blue) and 30% (magenta) of the signal to simulate a clear case of burst dynamics; a more realistic "burst" model in which we artificially inserted sporadic bursts of 0.17 dB to 10%, 20% or 30% of the signal to simulate recall-related activations similar to the response during individual recall-events reported in Figure 5; and finally, a 'baseline shift' model, in which we added a constant gain of 0.1 dB to 80%,90% or 100% of the signal. (b) Differences between the three models are evident when comparing amplitude changes across percentiles. As can be seen, only the 'baseline-shift' model leads to a constant gain across all percentiles. (c-d) In the actual data, category selective electrodes manifest statistical properties that are more compatible with the baseline-shift model: a significant amplitude gain across all percentiles during recall of the preferred category (* $p < 0.001$, preferred minus non-preferred category, signed-rank tests), with no significant inter-percentile differences ($p > 0.84$, Kruskal-Wallis test). For comparison, during stimuli presentation the same electrodes show an amplitude modulation more compatible with a "burst" model rather than baseline-shifts, as expected (yellow). Error bars represent SEM.

24. It is very unclear why the authors switch to a different contrast for the transient response analysis, compared to the baseline shift analysis (the transient response analysis contrasts preferred vs. non-preferred images; the baseline shift analysis contrasts responses to the preferred vs. non-preferred category). Perhaps the “preferred” image analysis is more sensitive to recall-triggered responses and/or it facilitates comparisons across regions? Regardless, the justification should be provided. It would also be useful to see the transient response to recalls from the preferred vs. non-preferred category, so this can be compared to the baseline-shift in those electrodes for the preferred vs. non-preferred category.

- This is an important point and we completely agree. We added a between category contrast in Figure 5a, which allows now a more straightforward comparison between the transient event-related responses and the baseline-shift results presented in Figure 3a. The complete event-related analysis is described in the revised results section as follows [line 227]:

“...we extracted the onset of each verbal report by analyzing the voice-recordings of each patient, following the previously published procedure³⁰ (see Methods). We then normalized the HFB signal by the geometric mean amplitude during resting-states and smoothed it using a 1,000 ms triangular window to compensate for uncertainties regarding the latencies of underlying neuronal events relative to speech onset. For each electrode we computed the event-related HFB response time-locked to the onset of each verbal report of recall, and separately averaged responses during recall of items from the ‘preferred’ and the ‘non-preferred’ category. Finally, we computed the grand-average responses across all category-selective electrodes in a time window of -5 to 7 seconds relative to speech onset. Recall events that were not separated by at least five seconds from the previous recollection were excluded from the analysis.

As shown in Figure 5a, in addition to a sustained amplitude increase during recall of items from the preferred category (as expected from the baseline-shift results described above), the recall moment was preceded by a transient differential activity: while recall of items from the preferred category involved a small amplitude increase, recall of non-preferred items involved a small amplitude decrease. A time-point by time-point, successive, two-tailed, paired t-test (blue line) comparing the preferred vs. the non-preferred category, indicated a quick rise in differential activity that anticipated the onset of verbal report by 1,500ms (significant time bins are marked in yellow, $P < 0.05$, cluster based correction using Monte Carlo simulation with 10,000 iterations in which the ‘preferred’ and ‘non-preferred’ labels were randomly shuffled over electrodes).

To increase sensitivity to recall-triggered responses and reinstatement of visual information regardless of category, we carried out a complementary analysis in which we contrasted in each electrode the responses during recall of 'preferred images' (top third of items that preferentially activated the electrode during viewing) and 'non-preferred images' (bottom third of items that least activated the electrode during viewing). As this contrast excludes all 'borderline' exemplars that activated the electrode in a partial manner, it should be more sensitive in detecting item-specific reactivation than the contrast of categories. Moreover, this contrast allowed comparing category selective electrodes to early visual electrodes (V1+V2), as both electrode groups manifested a clear preference to some exemplars during viewing (not necessarily related to a specific category). For distribution of 'preferred images' across exemplars and categories see Supplementary Figure 5.

The results of this analysis are depicted in Figure 5c-f, showing the grand-average recall responses in category selective electrodes (panel c-d) and early visual electrodes (panel e-f) for preferred and non-preferred images, in a time window of -5 to 7 seconds relative to speech onset. Here again, recall events that were not separated by at least five seconds from the previous recollection were excluded from the analysis. As can be seen, while high-order category selective electrodes showed a clear recall-triggered response, involving a small amplitude increase/decrease for preferred/non-preferred images respectively (see Supplementary fig. 5 for individual electrode data), early visual electrodes showed no significant response. To further test the generalization of this recall-triggered effect, we fitted the data with a mixed-effects model using the fixed factors of 'preference' (preferred/non-preferred) and 'time-bin' (baseline [-5 to -3 sec] / recall onset transient [-1 to 1 sec]), and the random factors of 'patient' and 'electrode'. We found a significant preference effect with higher activity levels for preferred images ($F(1,192)=13.71$, $P<0.001$), and a significant interaction between preference and time-bin ($F(1,192)=5.04$, $P=0.026$) (Fig. 5g)."

Figure 5. HFB response time-locked to the onset of individual recall events.

(a) Stimulus response in category selective electrodes to images from the preferred and non-preferred categories during the picture viewing stage. (b) Recall-related response in the same electrodes time-locked to the onset of individual recall events, showing both a sustained baseline-shift and a transient event-related component. The transient component involved both an

amplitude increase and decrease anticipating the recall of items from the preferred and non-preferred categories, respectively. The blue line presents t-values from a successive paired t-test comparing recall of preferred vs. non-preferred category. Significant time bins are indicated in yellow ($p < 0.05$, cluster-based correction using Monte Carlo simulation). Shaded areas reflect unbiased within-electrode corrected 95% confidence interval³⁵. **(c-d)** an alternative contrast applied to the same electrodes showing stimulus response of preferred (top third responses) and non-preferred (bottom third responses) images during the picture viewing stage, and the corresponding responses during recall (same notations as in panel b). Here again a transient differential activity component appeared to anticipate the recall event by ~1.5 seconds. **(e-f)** Applying the same contrast (i.e. preferred vs. non-preferred images) in early visual electrodes (V1+V2) did not reveal any significant recall-related response. **(g)** Comparing recall onset response in category selective electrodes to the immediate pre-recall baseline using a mixed-effects model revealed a significant interaction between preference and time-bin ($F(1,192)=5.04$, $*p < 0.05$), further supporting the observation that HFB amplitude was increased/decreased during recall of preferred/non-preferred images, respectively. **(h)** Anatomical location of face-selective (red), place-selective (green), V1 (blue) and V2 (light-blue) electrodes on an inflated template brain.

25. Many of the statistics in this paper appear to ignore the fact that different participants contributed different numbers of electrodes (i.e., they make no distinction between electrodes from the same vs. different participants). This approach is not ideal because it does not distinguish between within-subject and between-subject variance. It would be better to run mixed-effects analyses that distinguish between these types of variance. Alternatively, the authors could run a subject-level bootstrap analysis (i.e., resample participants with replacement) to verify that the effect is reliable at the population level. If the authors run a bootstrap, they should also try a variant where they z-score these measures within-subject – if this analysis works, the authors can conclude that the effect is present within individual subjects (as opposed to being driven, e.g., by some participants having electrodes that are place selective at encoding and retrieval, and other participants having electrodes that are face selective at encoding and retrieval).

- We thank the reviewer for pointing out this issue. To allow better generalization of the results we replaced the repeated measures ANOVA with a mixed-effects analysis, in which ‘patient’ and ‘electrode’ were statistically treated as random factors. Thus, for example, in figure 3a, median amplitude values were fitted with a two-way mixed-effects model using ‘electrode group’ (face-selective/place-selective) and ‘recalled category’ (faces/places) as fixed factors, and ‘patient’ and ‘electrode’ as random factors [line 116]. Further details about the statistical method can be found in the revised Methods section [line 730]:

“Mixed-effects analyses were carried out using the LME4 package³⁶ implemented in R³⁷. HFB amplitude values (dB) were fitted with a random intercept model formulated as follows: $Y \sim X1 * X2 * X3 + (1 | Patient / Electrode)$, where X1, X2, etc. are the relevant fixed factors in each analysis, and the terms in parentheses are the nested random factors of ‘Patient’ and ‘Electrode’. Main effects and interactions were tested using the afex R package³⁸. Degrees of freedom were computed using the

Kenward-Roger (KR) method. Post-hoc comparisons were carried out using lsmeans R package³⁹ and were corrected for multiple comparisons using FDR⁴⁰ ($P < 0.05$). Although including random slopes when applicable is generally recommended⁴¹, these could not be included for our dataset since they led to over-parameterization (model unidentifiability).”

26. Why did the authors choose to normalize by the geometric mean HFB amplitude during rest (as opposed to, say, the arithmetic mean)? Does this choice matter?

- For clarification, we computed the mean resting state amplitude after transformation to decibel units ($10 \times \log_{10}$) – thus the averaging was done on a log scale which is equivalent to taking the geometric mean.
- Normalizing the signal by the geometric rather than the arithmetic mean did not affect any of the main results. It was simply more convenient for analysis purposes.
- In general, since the HFB signal tend to follow a log-normal distribution (similar to other measures of population firing rates⁴²) – it makes sense to use the geometric mean as a central tendency, as it coincides with the median and is less affected by extreme values.

REFERENCES

1. Nir, Y. *et al.* Coupling between Neuronal Firing Rate, Gamma LFP, and BOLD fMRI Is Related to Interneuronal Correlations. *Curr. Biol.* **17**, 1275–1285 (2007).
2. Miller, K. J. *et al.* Broadband changes in the cortical surface potential track activation of functionally diverse neuronal populations. *Neuroimage* **85**, 711–720 (2014).
3. Mukamel, R. Coupling Between Neuronal Firing, Field Potentials, and fMRI in Human Auditory Cortex. *Science* (80-.). **309**, 951–954 (2005).
4. Tulving, E. & Pearlstone, Z. Availability versus accessibility of information in memory for words. *J. Verbal Learning Verbal Behav.* **5**, 381–391 (1966).
5. Kahana, M. J., Dolan, E. D., Sauder, C. L. & Wingfield, A. Intrusions in episodic recall: age differences in editing of overt responses. *J. Gerontol. B. Psychol. Sci. Soc. Sci.* **60**, P92–P97 (2005).
6. Jacques, C. *et al.* Corresponding ECoG and fMRI category-selective signals in human ventral temporal cortex. *Neuropsychologia* **83**, 14–28 (2016).
7. Grill-Spector, K. & Weiner, K. S. The functional architecture of the ventral temporal cortex and its role in categorization. *Nat. Rev. Neurosci.* **15**, 536–548 (2014).
8. Noy, N. *et al.* Ignition’s glow: Ultra-fast spread of global cortical activity accompanying local ‘ignitions’ in visual cortex during conscious visual perception. *Conscious. Cogn.* **35**, 206–224 (2015).
9. Fletcher, P. C. Frontal lobes and human memory: Insights from functional neuroimaging. *Brain* **124**, 849–881 (2001).
10. Fletcher, P. C., Shallice, T., Frith, C. D., Frackowiak, R. S. J. & Dolan, R. J. The functional roles of prefrontal cortex in episodic memory. II. Retrieval. *Brain* **121**, 1249–1256 (1998).
11. Badre, D. & Wagner, A. D. Left ventrolateral prefrontal cortex and the cognitive control of memory. *Neuropsychologia* **45**, 2883–2901 (2007).
12. Dobbins, I. G. & Wagner, A. D. Domain-general and domain-sensitive prefrontal mechanisms for recollecting events and detecting novelty. *Cereb. Cortex* **15**, 1768–1778 (2005).
13. Velanova, K. *et al.* Functional-anatomic correlates of sustained and transient processing components engaged during controlled retrieval. *J. Neurosci.* **23**, 8460–70 (2003).
14. Long, N. M., Oztekin, I. & Badre, D. Separable Prefrontal Cortex Contributions to Free Recall. *J. Neurosci.* **30**, 10967–10976 (2010).
15. Ciaramelli, E., Grady, C., Levine, B., Ween, J. & Moscovitch, M. Top-Down and Bottom-Up Attention to Memory Are Dissociated in Posterior Parietal Cortex: Neuroimaging and Neuropsychological Evidence. *J. Neurosci.* **30**, 4943–4956 (2010).
16. Cabeza, R., Ciaramelli, E., Olson, I. R. & Moscovitch, M. The parietal cortex and episodic memory: an attentional

- account. *Nat. Rev. Neurosci.* **9**, 613–25 (2008).
17. Ciaramelli, E., Grady, C. L. & Moscovitch, M. Top-down and bottom-up attention to memory: A hypothesis (AtoM) on the role of the posterior parietal cortex in memory retrieval. *Neuropsychologia* **46**, 1828–1851 (2008).
 18. Wagner, A. D., Shannon, B. J., Kahn, I. & Buckner, R. L. Parietal lobe contributions to episodic memory retrieval. *Trends Cogn. Sci.* **9**, 445–453 (2005).
 19. Moscovitch, M., Cabeza, R., Winocur, G. & Nadel, L. Episodic memory and beyond: The hippocampus and neocortex in transformation. *Annu. Rev. Psychol.* **67**, 105–134 (2016).
 20. Moscovitch, M. The hippocampus as a ‘stupid,’ domain-specific module: Implications for theories of recent and remote memory, and of imagination. *Can. J. Exp. Psychol. Can. Psychol. expérimentale* **62**, 62–79 (2008).
 21. Badre, D., Poldrack, R. A., Pare-Blagoev, E. J., Insler, R. Z. & Wagner, A. D. Dissociable Controlled Retrieval and Generalized Selection Mechanisms in Ventrolateral Prefrontal Cortex. *Neuron* **47**, 907–918 (2005).
 22. Tanaka, K. Z. *et al.* Cortical Representations Are Reinstated by the Hippocampus during Memory Retrieval. *Neuron* **84**, 347–354 (2014).
 23. Rothschild, G., Eban, E. & Frank, L. M. A cortical–hippocampal–cortical loop of information processing during memory consolidation. *Nat. Neurosci.* **20**, 251–259 (2016).
 24. Rugg, M. D. & Vilberg, K. L. Brain networks underlying episodic memory retrieval. *Curr. Opin. Neurobiol.* **23**, 255–260 (2013).
 25. Morton, N. W. *et al.* Category-specific neural oscillations predict recall organization during memory search. *Cereb. Cortex* **23**, 2407–2422 (2013).
 26. Becker, S. & Lim, J. A computational model of prefrontal control in free recall: strategic memory use in the California Verbal Learning Task. *J. Cogn. Neurosci.* **15**, 821–832 (2003).
 27. Raaijmakers, J. G. & Shiffrin, R. M. Search of associative memory. *Psychol. Rev.* **88**, 93–134 (1981).
 28. Polyn, S. M., Norman, K. a & Kahana, M. J. A context maintenance and retrieval model of organizational processes in free recall. *Psychol. Rev.* **116**, 129–156 (2009).
 29. Gronlund, S. D. & Shiffrin, R. M. Retrieval Strategies in Recall of Natural Categories and Categorized Lists. *J. Exp. Psychol. Learn. Mem. Cogn.* **12**, 550–561 (1986).
 30. Gelbard-Sagiv, H., Mukamel, R., Harel, M., Malach, R. & Fried, I. Internally generated reactivation of single neurons in human hippocampus during free recall. *Science (80-.)*. **322**, 96–101 (2008).
 31. Hesselmann, G., Kell, C. a, Eger, E. & Kleinschmidt, A. Spontaneous local variations in ongoing neural activity bias perceptual decisions. *Proc. Natl. Acad. Sci. U. S. A.* **105**, 10984–10989 (2008).
 32. Schurger, a., Sitt, J. D. & Dehaene, S. PNAS Plus: An accumulator model for spontaneous neural activity prior to self-initiated movement. *Proc. Natl. Acad. Sci.* **109**, E2904–E2913 (2012).
 33. Moutard, C., Dehaene, S. & Malach, R. Spontaneous Fluctuations and Non-linear Ignitions: Two Dynamic Faces of

- Cortical Recurrent Loops. *Neuron* **88**, 194–206 (2015).
34. Nir, Y. *et al.* Interhemispheric correlations of slow spontaneous neuronal fluctuations revealed in human sensory cortex. *Nat. Neurosci.* **11**, 1100–1108 (2008).
 35. Morey, R. D. Confidence Intervals from Normalized Data: A correction to Cousineau (2005). *Tutor. Quant. Methods Psychol.* **4**, 61–64 (2008).
 36. Bates, D., Maechler, M., Bolker, B., Walker, S. & others. lme4: Linear mixed-effects models using Eigen and S4. *R Packag. version 1*, (2014).
 37. Team, R. C. R: A Language and Environment for Statistical Computing. Vienna: R Foundation for Statistical Computing. Available online at: <http://www.R-project.org> (2014).
 38. Singmann, H., Bolker, B. & Westfall, J. afex: Analysis of factorial experiments. *R Packag. version 0.13--145* (2015).
 39. Lenth, R. V & others. Least-squares means: the R package lsmeans. *J Stat Softw* **69**, 1–33 (2016).
 40. Benjamini, Y. & Hochberg, Y. Controlling the false discovery rate: a practical and powerful approach to multiple testing. *J. R. Stat. Soc. B* **57**, 289–300 (1995).
 41. Bates, D., Mächler, M., Bolker, B. & Walker, S. Fitting Linear Mixed-Effects Models Using **lme4**. *J. Stat. Softw.* **67**, (2015).
 42. Buzsáki, G. & Mizuseki, K. The log-dynamic brain: how skewed distributions affect network operations. *Nat. Rev. Neurosci.* **15**, 264–78 (2014).

REVIEWERS' COMMENTS:

Reviewer #1 (Remarks to the Author):

I have read the revision and thought that the authors did an excellent job in addressing my comments and, I believe, the comments of the other reviewers. In doing so, they often conducted additional analyses and modified their paper accordingly. In short, I am happy with the revision and have no more substantial comments (but see some minor comments below). I think the paper makes an important contribution to the literature.

Minor comments

Some of the ideas discussed in this paper are related to Tulving's notions of retrieval mode, yet there is no citation to that work. It would be worthwhile to include it (around page 3, line 50, and possibly in the discussion).

The transient signal in parietal cortex has been noted by other investigators, most especially Michael Rugg. A reference to his studies on transient and sustained signals in parietal cortex is appropriate.

Likewise, Bryce Kuhl has shown that there is category specificity in parietal cortex. Referring to his work is also appropriate.

P 8, line 168. Insert a comma before "however"

P 9, line 205. Does Bursts' need the apostrophe?

Reviewer #2 (Remarks to the Author):

The authors have provided a clear and detailed response, I have no further comment.

Reviewer #3 (Remarks to the Author):

Norman and colleagues present an important study of category-specific brain responses during targeting of categorized materials. The authors were very responsive to the comments of the reviewers, and have greatly improved the manuscript. They carried out a number of important control analyses, including re-doing the main analysis excluding all vocalizations, and including an analysis of neural response preceding recall intrusions, and relative to a prompt by the experimenter. Taken together, the authors present a comprehensive analysis of a rich set of neural dynamics. Characterizing the neural basis of category targeting during memory search is highly significant, and will be of broad interest to neuroscientists and psychologists. The results have important implications for theories of executive control, episodic and semantic memory.

In response to reviewer 1, the authors report that 10 percent of all responses were from the wrong category, and suggest that this number is in line with previous reports in the literature. But it really isn't in line with the previous literature at all. The authors cite a study by Tulving and Pearlstone, where 949 people were tested using a variety of category cuing techniques. T&P report that there were 73 noncategorical intrusions in all the 949 recall protocols, but each recall protocol involved many responses, it looks like 10-30 per person, which means that really the likelihood of an extra-category intrusion was less than 1 percent. T&P were trying to say that noncategorical intrusions were exceedingly rare! (and they say so explicitly on pg 386 of the 1966 paper) The other citation, Kahana et al 2005, looks at intrusions in free recall, but these were non-categorized lists, just random words from a large pool of words. The Kahana paper isn't relevant here, there is no category structure in that paper.

I think the authors would be better served by acknowledging that this intrusion rate is high relative to reports in the literature.

In response to point 10 of reviewer 1, the authors say that patients were instructed to "verbally report on everything that came to mind during the free recall period". This detail is not reported in the Methods section, but it is critical here. The Methods says that the patient was asked to report items from a specific category, but if they were also told to say everything that comes to mind, then this clearly explains why they are making 10% intrusions. Don't say that this is a normal level of intrusions if you explicitly told the patients that saying things from the other category is ok, just be more explicit about the task instructions!

Anyway, I just want to make sure these citations aren't used to suggest that this is a normal intrusion rate for recall from a particular category. And the important thing is that in the end, the authors turn this intrusion rate into a strength of the paper, by showing that the neural response prior to intrusions shows a loss of category-specificity.

Minor comments:

Page 20, line 459, looking at the long anticipatory buildup prior to recall response, a similar thing was observed by Polyn et al. (2005) in fMRI.

Page 21, line 503, this should say Figure 10.

Reviewer #4 (Remarks to the Author):

The authors have fully addressed my concerns, and have also added some nice new analyses in response to the other reviewers.

Sincerely,

Ken Norman (I sign all of my reviews)

Reviewer #1:

1. Some of the ideas discussed in this paper are related to Tulving's notions of retrieval mode, yet there is no citation to that work. It would be worthwhile to include it (around page 3, line 50, and possibly in the discussion).
 - We thank the reviewer for drawing our attention to this issue. We now cite Tulving's work in the last paragraph of the introduction and in the first paragraph of the discussion:

“...the idea of an internally maintained cue that acts to bias retrieval competition in favor of specific classes or categories, an idea which is widespread in the cognitive modeling memory literature¹⁻⁴ and plays an essential part in Tulving's notions of retrieval mode⁵.”
2. The transient signal in parietal cortex has been noted by other investigators, most especially Michael Rugg. A reference to his studies on transient and sustained signals in parietal cortex is appropriate. Likewise, Bryce Kuhl has shown that there is category specificity in parietal cortex. Referring to his work is also appropriate.
 - We included references to Rugg's and Kuhl's works in the appropriate places in the discussion:

“The persistent activity in VPC throughout the verbal report may suggest an active role in maintaining an internal representation of the retrieved material⁶⁻⁸.”
3. P8, line 168. Insert a comma before "however"; P9, line 205. Does Bursts' need the apostrophe?
 - Fixed.

Reviewer #3:

4. ...In response to point 10 of reviewer 1, the authors say that patients were instructed to "verbally report on everything that came to mind during the free recall period". This detail is not reported in the Methods section, but it is critical here. The Methods says that the patient was asked to report items from a specific category, but if they were also told to say everything that comes to mind, then this clearly explains why they are making 10% intrusions. Don't say that this is a normal level of intrusions if you explicitly told the patients that saying things from the other category is ok, just be more explicit about the task instructions.
 - We thank the reviewer for noting this issue; we indeed overlooked this discrepancy. We have corrected the relevant parts and included the following sentence in the second paragraph of the results:

“Such intrusion rate is considered high compared to previous reports⁹, however, this discrepancy most likely reflects the fact that unlike previous studies, here we specifically instructed the patients to report on everything that comes to mind during the free recall period.

- We have also added the following sentence in the Methods section:

“The instructions also emphasized to report on everything that came to mind during the free recall period.”

5. Page 20, line 459, looking at the long anticipatory buildup prior to recall response, a similar thing was observed by Polyn et al. (2005) in fMRI.

- We added a reference to Polyn’s study in the relevant paragraph in the discussion.

6. Page 21, line 503, this should say Figure 10.

- Fixed.

References

1. Becker, S. & Lim, J. A computational model of prefrontal control in free recall: strategic memory use in the California Verbal Learning Task. *J. Cogn. Neurosci.* **15**, 821–832 (2003).
2. Raaijmakers, J. G. & Shiffrin, R. M. Search of associative memory. *Psychol. Rev.* **88**, 93–134 (1981).
3. Polyn, S. M., Norman, K. a & Kahana, M. J. A context maintenance and retrieval model of organizational processes in free recall. *Psychol. Rev.* **116**, 129–156 (2009).
4. Gronlund, S. D. & Shiffrin, R. M. Retrieval Strategies in Recall of Natural Categories and Categorized Lists. *J. Exp. Psychol. Learn. Mem. Cogn.* **12**, 550–561 (1986).
5. Tulving, E. Elements of episodic memory. (1985).
6. Wagner, A. D., Shannon, B. J., Kahn, I. & Buckner, R. L. Parietal lobe contributions to episodic memory retrieval. *Trends Cogn. Sci.* **9**, 445–453 (2005).
7. Vilberg, K. L. & Rugg, M. D. The neural correlates of recollection: Transient versus sustained fMRI effects. *J. Neurosci.* **32**, 15679–15687 (2012).
8. Kuhl, B. A. & Chun, M. M. Successful remembering elicits event-specific activity patterns in lateral parietal cortex. *J Neurosci* **34**, 8051–8060 (2014).
9. Tulving, E. & Pearlstone, Z. Availability versus accessibility of information in memory for words. *J. Verbal Learning Verbal Behav.* **5**, 381–391 (1966).